# Testing ion exchange resin for quantifying bulk and throughfall deposition of macro- and micro-elements in forests

**Marleen A. E. Vos**[1]**, Wim de Vries**[2]**, G. F. (Ciska) Veen**[3]**, Marcel R. Hoosbeek**[4]**, and Frank J. Sterck**[1]

[1]Forest Ecology and Forest Management Group, Wageningen University and Research,
Wageningen 6700 AA, the Netherlands
[2]Earth Systems and Global Change Group, Wageningen University and Research, Wageningen 6700 AA, the Netherlands
[3]Department of Terrestrial Ecology, Netherlands Institute of Ecology (NIOO-KNAW), Wageningen 6708 PB, the Netherlands
[4]Soil Chemistry Group, Wageningen University and Research, Wageningen 6700 AA, the Netherlands

**Correspondence:** Marleen A. E. Vos (marleen.vos@wur.nl)

**Abstract.** Atmospheric deposition is a major nutrient influx in ecosystems, while high anthropogenic deposition may disrupt ecosystem functioning. Quantification of the deposition flux is required to understand the impact of such anthropogenic pollution. However, current methods to measure nutrient deposition are costly, labor-intensive and potentially inaccurate.

Ion exchange resin (IER) appears to be a promising cost- and labor-effective method. The IER method is potentially suited for deposition measurements on coarse timescales and for areas with little rainfall and/or low elemental concentrations. The accuracy of the IER method is, however, hardly classified beyond nitrogen. We tested the IER method for bulk deposition and throughfall measurements of macro- and micro-elements, assessing resin adsorption capacity, recovery efficiency and field behavior.

We show that IER is able to adsorb 100 % of Ca, Cu, Fe, K, Mg, Mn, P, S, Zn and $NO_3^-$ and > 96 % of P and Na. Loading the resin beyond its capacity resulted mainly in losses of Na, P and $NH_4^+$, while losses of Ca, Cu, Fe, Mg, Mn and Zn were hardly detected. Heat (40 °C), drought and frost ($-15$ °C) reduced the adsorption of P by 25 %. Recovery was close to 100 % for $NH_4^+$ and $NO_3^-$ using KCl solution (1 or 2 M), while high (83 %–93 %) recoveries of Ca, Cu, Fe, K, Mg, Mn and S were found using HCl as an extractant (2–4 M). We found good agreement between the conventional method and the IER method for field conditions.

Overall, IER is a powerful tool for the measurement of atmospheric deposition of a broad range of elements as the measurements showed high accuracy. The IER method therefore has the potential to expand current monitoring networks and increase the number of sampling sites.

## 1 Introduction

Atmospheric deposition is a major nutrient influx in many ecosystems and therefore crucial for ecosystem functioning (Van Langenhove et al., 2020). However, due to anthropogenic pollution, atmospheric deposition can potentially disrupt ecosystem nutrient balances, leading to exceedance of critical deposition thresholds of, for example, nitrogen, which can in turn degrade ecosystem functioning (de Vries et al., 2011; Rabalais, 2002). Such degradation of ecosystems involves accelerated soil acidification and reduced availability of critical soil nutrients, such as base cations, which has detrimental impacts on biodiversity and water quality (Stevens et al., 2004; Houdijk et al., 1993; Solberg et al., 2009; Lu et al., 2014; Johansson et al., 2001; Bowman et al., 2008; Horswill et al., 2008; de Vries et al., 2014). Atmospheric deposition is therefore of major importance to many ecosystems, and monitoring deposition is necessary for policy, management and conservation.

Measurements of atmospheric deposition are, however, costly and labor-intensive. Direct measurements of dry deposition (i.e., input of elements as airborne particles) and wet-only deposition (i.e., input of elements via precipitation) (Balestrini et al., 2007; Lovett and Reiners, 1986) are scarce,

and current technology limits widespread measurements. For forests, the common method to assess total deposition (i.e., wet and dry deposition combined) is the collection of precipitation below forests, called throughfall, in collection devices of various shapes and sizes, while accounting for canopy exchange (Thimonier, 1998; Draaijers et al., 1996), which is based on the additional measurement of precipitation outside the forest, known as bulk deposition. The combined measurement of nutrient inputs in precipitation in and outside forests, further called the bulk deposition method in this paper, is readopted in many monitoring networks (i.e., ICP forest network (de Vries et al., 2003; De Vries et al., 2007; Bleeker et al., 2003), the DONAIRE network (Pey et al., 2020) and the nationwide monitoring network in China (Xu et al., 2019)). However, the use of bulk deposition measurements requires frequent (up to weekly) sampling as $NH_4^+$ in the collected rainwater may relatively rapidly be transformed to $NO_3^-$ by nitrification, with the speed being dependent on local weather conditions (Clarke et al., 2016). The high sampling frequency and the high cost of traveling and laboratory analysis limit the spatial and temporal scales at which this method can be applied. The alternative is larger sampling intervals, but this may cause inaccurate assessment of the input, especially of $NH_4^+$ versus $NO_3^-$. An adequate assessment of both N compounds is especially crucial in regions where the allocation of N sources is highly sensitive ($NH_4^+$ being caused by $NH_3$ emissions from agriculture and $NO_3^-$ from $NO_x$ emissions by traffic and industry). Better alternatives are needed to measure deposition efficiently in the field, improve the reliability of the measurements, reduce sampling effort and costs, and thus allow for more effective large-scale deposition monitoring programs.

The ion exchange resin method (IER) was previously developed to measure bulk deposition at large spatial and temporal scales, but use of the method is yet limited to remote areas (Brumbaugh et al., 2016), the monitoring network of California (Fenn et al., 2018) or case studies (Hoffman et al., 2019; Garcia-Gomez et al., 2016; Clow et al., 2015). Widespread application of the IER method is promising as the method allows the accumulated deposition to be measured over long time periods (up to a year), which strongly reduces both the sampling effort in the field and the number of lab analysis, leading to major cost savings (Fenn and Poth, 2004; Kohler et al., 2012). Furthermore, the method is more reliable for nitrogen, as the resin likely inhibits mineralization, nitrification and denitrification, which can be affected by local weather conditions, as discussed by Fenn and Poth (2004) and Kohler et al. (2012). Finally, the IER method is able to measure the deposition in areas with low rainfall or low elemental concentrations, avoiding problems with the detection limit and minimal sample size required in the bulk deposition method (Kohler et al., 2012).

The IER method is most commonly used for $NH_4^+$ and $NO_3^-$ measurements (Fang et al., 2011; Fenn et al., 2002; Fenn and Poth, 2004; Kohler et al., 2012; Clow et al., 2015; Garcia-Gomez et al., 2016; Hoffman et al., 2019), but few studies reported measurements of other elements (e.g., S, K, Ca, Mg, Na and Cl) (Simkin et al., 2004; Van Dam et al., 1987; Fenn et al., 2018). The applicability of the method to measure a broad range of elements depends on the performance of the resin, measured as the adsorption capacity (percentage of the total element flux bound to the resin) and the recovery efficiency of elements (percentage of the total element flux recovered from the resin) (Garcia-Gomez et al., 2016). Often though, the adsorption capacity and recovery efficiencies are not reported (Risch et al., 2020; Boutin et al., 2015; Fenn and Poth, 2004; Fenn et al., 2015). Studies reporting the adsorption capacity (Fang et al., 2011; Simkin et al., 2004; Garcia-Gomez et al., 2016) only describe the adsorption of a limited number of elements under laboratory conditions. Recovery efficiency under laboratory conditions is more often reported, with high recovery efficiencies in general (87 %–100 %), although the recovery of some macro-elements (i.e., Ca and Mg) was below 50 % (Simkin et al., 2004; Wieder et al., 2016; Clow et al., 2015; Cerón et al., 2017; Kohler et al., 2012; Fang et al., 2011). Despite the promising applicability, the adequacy of the IER method to derive bulk deposition and throughfall under field conditions is hardly tested. The limited information on adsorption capacity combined with bad recoveries for some elements (i.e., Ca, Mg, Fe and Al) potentially limits the use of the IER method for bulk deposition measurements.

The adsorption capacity and the recovery efficiency can be influenced by environmental field conditions like drought, frost or high temperatures (Qian and Schoenau, 2002; Bayar et al., 2012). However, there are hardly any studies testing the influence of environmental field conditions on both the adsorption capacity and the recovery efficiency of the resin. Furthermore, most tests refer to bulk deposition, whereas atmospheric deposition on forest is also measured as throughfall underneath vegetation canopies. Dissolved organic substances are higher in throughfall than in bulk deposition for which the adsorption capacity of the resin is lower (Langlois et al., 2003). Overall, recovery rates from resin exposed to environmental field conditions appear to be lower, urging the need for better evaluation of IER performance under field conditions (Krupa and Legge, 2000; Brumbaugh et al., 2016). Therefore, new tests are necessary to evaluate the effect of environmental conditions and organically rich throughfall on the elemental recovery from the resin.

The recovery efficiency can be optimized by the use of different extraction methods. An often used extraction method is 2 M KCl for nitrogen extraction (Hoffman et al., 2019; Fenn et al., 2002; Garcia-Gomez et al., 2016), but combinations of either KI, $HNO_3$, NaCl or $H_2SO_4$ and HCl were also used (Fenn et al., 2018; Brumbaugh et al., 2016; Kohler et al., 2012; Van Dam et al., 1991). The KCl extraction method and the KI extraction method do not allow measurements of K deposition and are, as high dissolved salt solutions, problematic for measurements using an inductively coupled plasma–

atomic emission spectrometer (ICP-AES) (Brumbaugh et al., 2016; Hislop and Hornbeck, 2002). New tests are therefore needed to increase the recovery efficiency, allowing a broad range of elements to be measured, including all macronutri-ents and micronutrients.

In this study we aim to test the capacity of IER as a method to quantify atmospheric deposition for a broad range of macro- and micro-elements, comparing results under laboratory and field conditions and in the latter case comparing bulk deposition and throughfall. We first tested the method under controlled laboratory conditions to indicate efficient resin volumes and to assess adsorption capacities and recovery efficiencies. Next, the behavior of the IER method was tested under field conditions, covering a gradient from closed forests to open areas, to account for the effect of dissolved organic substances on the performance of the resin columns. From this, we provide different methodological protocols with accuracies for detecting different macro- and micro-elements under field conditions, including forests.

## 2 Methods

### 2.1 Preparation of the resin columns

We prepared 45 resin columns for the laboratory tests of elemental adsorption and recovery (including the blanks), followed by the preparation of 30 columns for the field test of the IER method. The resin columns had a volume of 15.7 mL and an inner diameter and length of 12.4 and 130 mm, respectively. First, the empty resin columns were cleaned using 0.2 M HCl and demineralized water to remove weakly attached chemicals from the column walls. Then, the cleaned and dried resin columns were washed three times with demineralized water in the laboratory prior to filling the column with IER to ensure the removal of any remaining HCl. To fill the columns, a plug of clean polyester fiber was placed inside the resin column and pushed to the bottom. A cleaned cap was screwed loosely at the bottom of the resin column, stabilizing the polyester plug. The resin column was placed vertically above a container to collect the reagents. The ion exchange resin (Amberlite IRN-150, a 1 : 1 mixture of $H^+$ and $OH^-$) was washed with 8 L demineralized water in batches of 500 g of resin to remove small particles within the resin and to remove the resin's smell, which could attract animals. All liquids were drained from the resin using a vacuum pump, and for each resin column, 9.8 g of resin was weighed out and poured into the resin column using a pipette with demineralized water. When excess water had passed through, a second plug of polyester fiber was placed on top of the resin, and both sides of the column were screwed tightly with cleaned caps. A schematic overview of these steps is in Fig. 1.

### 2.2 Laboratory tests

The adsorption capacity and the recovery efficiency of the IER (Amberlite IRN 150 $H^+$ and $OH^-$ form) were tested at the Soil Chemistry Laboratory (CBLB), Wageningen University. First, based on existing wet-deposition data from nearby measurement stations located in Biest-Houtakker, Speuld, De Zilk and Vredepeel (NL) (RIVM, 2015), we estimated the bulk deposition amounts (kg ha$^{-1}$) for different elements and then used those to determine the needed molarity of the solution that was used to test the adsorption capacity of the resin. Both the adsorption capacity and recovery efficiency were subsequently tested for annual maximum bulk deposition rates across the Netherlands of the following elements: $PO_4^{2-}$, $SO_4^{2-}$, $N–NO_2^- + N–NO_3^-$, $N–NH_4^+$, $Ca^{2+}$, $Mg^{2+}$, $K^+$, $Na^+$, $Fe^{2+}$, $Mn^{2+}$, $Cu^{2+}$ and $Zn^{2+}$.

To estimate the maximum bulk deposition values, the monthly measurements of existing bulk deposition data of the nearby weather stations (µmol L$^{-1}$) (RIVM, 2015), were summed to seasonal concentrations, expressed in mg L$^{-1}$. Then, the stations were selected with the highest seasonal deposition, occurring during summer, for both macro- and micronutrients, based on the total molarity of the rainwater. These seasonal concentrations were then multiplied by the precipitation (in L) that would be captured by the funnel, by multiplying the recorded precipitation (in mm or L m$^{-2}$) by the horizontal surface of the funnel (in m$^2$) to estimate the total deposition captured by a funnel. Then, the deposition of the summer was multiplied by 2, which is an average multiplication factor to convert bulk deposition to throughfall. This average multiplication factor is based on the reported values for the ratio of throughfall to bulk deposition of the tracer Na (Table S1 in the Supplement). The total elemental content of this throughfall flux, multiplied by 4 (assuming that the summer values are representative of the entire year, which is a precautionary approach), was dissolved in a 1 L solution separately for macro- and micronutrients using stock solutions, resulting in an extraction solution containing values reflecting the maximum annual total deposition in the Netherlands (Table 1).

The adsorption capacity (i.e., percentage of total elemental influx adsorbed by the resin) was tested using 18 resin columns for laboratory tests. Out of these 18 columns, 9 columns were used to mimic heat, drought and frost conditions, and 9 columns were used to test the column's capacity (Table S2). Heat, drought and frost conditions were mimicked using 3 columns for each treatment, which consisted of heating to 40 °C, drying at 20 °C to a constant weight and freezing at $-19$ °C for 72 h, respectively, followed by dripwise loading with the macro- and micro-solution. The resin's capacity was simulated by dripping the macro- and micro-solutions through the resin columns using the normal concentration (3 columns and for the heat, drought and frost conditions), the double concentration (3 columns) and the triple concentration (3 columns), loading the columns up to

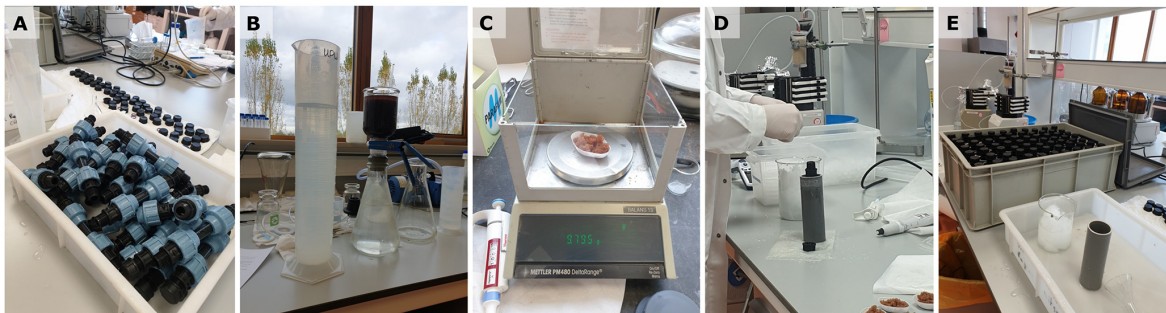

**Figure 1.** Preparation of the resin columns. A: cleaned resin columns prior to filling with IER. B: cleaning of the Amberlite IRN-150 exchange resin using a vacuum pump. C: weighing the resin prior to filling the resin column. D: resin column stabilized in a holder during filling with resin. E: overview of filled resin columns with the resin column stabilizer and the polyester plugs shown in the front.

**Table 1.** The throughfall flux used to test the adsorption capacity and recovery efficiency of the ion exchange resin. We used stock solutions with known molarity (M) to make the macro- and micro-solution used to drip through the resin. The total volume of the used stock solution (in $mL\,L^{-1}$) and the concentration (in µmol) per element are given.

| Stock solution | | Type | Total | Ca | Cu | Cl | Fe | K | Mg | Mn | Na | $PO_4$ | $SO_4$ | Zn | $NH_4^+$ | $NO_3^-$ |
|---|---|---|---|---|---|---|---|---|---|---|---|---|---|---|---|---|
| Code | M | | $mL\,L^{-1}$ | | | | | | | µmol | | | | | | |
| $Na_2SO_4$ | 0.5 | Macro | 0.90 | | | | | | | | 450 | | 450 | | | |
| NaCl | 1 | Macro | 1.40 | | | 1400 | | | | | 1400 | | | | | |
| $KNO_3$ | 1 | Macro | 0.18 | | | | | 180 | | | | | | | | 180 |
| $KH_2PO_4$ | 1 | Macro | 0.02 | | | | | 20 | | | | 20 | | | | |
| $NH_4NO_3$ | 1 | Macro | 1.82 | | | | | | | | | | | | 1820 | 1820 |
| $NH_4Cl$ | 1 | Macro | 2.18 | | | 2180 | | | | | | | | | 2180 | |
| $MgSO_4$ | 1 | Macro | 0.3 | | | | | | 300 | | | | 300 | | | |
| $CaCl_2$ | 0.5 | Macro | 0.8 | 400 | | 400 | | | | | | | | | | |
| $FeCl_2$ | 0.1 | Micro | 6.0 | | | 600 | 600 | | | | | | | | | |
| $Cu(NO_3)_2.3H_2O$ | 0.275 | Micro | 0.036 | | 9.9 | | | | | | | | | | | 9.9 |
| $Zn(NO_3)_2.6H_2O$ | 0.267 | Micro | 0.075 | | | | | | | | | | | 20 | | 20 |
| $Mn.SO_4.H_2O$ | 0.01 | Micro | 15.0 | | | | | | | 150 | | | 150 | | | |
| | | | Total | 400 | 9.9 | 4580 | 600 | 200 | 300 | 150 | 1850 | 20 | 900 | 20 | 4000 | 2030 |

70 %, 140 % and 210 % of their capacity, respectively. The exchange capacity of the resin as reported by the manufacturer was $\geq 0.6\,mol\,L^{-1}$ for the anion bed and $\geq 0.7\,mol\,L^{-1}$ for the cation bed. Samples of the leachate were taken when all the solution was drained from the resin (after approximately 4 h). Three resin columns loaded up to 70 % of the resin's exchange capacity were thereafter flushed with demineralized water to test the stability of the adsorption. This stability needed to be tested to check whether the ion exchange resin would release nutrients when exposed to (very) wet conditions. Furthermore, demineralized water used to clean the resin was taken as a blank sample for the adsorption test.

The recovery efficiency (i.e., percentage of total elemental flux recovered from the resin) was tested using 36 loaded resin columns in laboratory tests. In addition to these 36 loaded columns, two blanks were included to distinguish between the recovery efficiency of the loaded solution and background contamination from the resin or contamination caused by sample handling in the laboratory. Only the columns loaded with the double and triple concentration of the macro- and micro-solutions were excluded (Table S2). All unloaded columns were, similar to the previously loaded columns, drip-wise-loaded with the macro- and micro-solutions. Recovery efficiency was tested using a 2 M KCl extraction for $NH_4^+$ and $NO_3^-$ based on previous reported high recovery rates (Fenn et al., 2002; Fenn and Poth, 2004; Fang et al., 2011; Kohler et al., 2012; Clow et al., 2015; Hoffman et al., 2019) and multiple molarities of HCl (ranging from 1 to 4 M) for the other elements (Ca, Mg, K, Fe, Mn, Zn, Cu, Na and S) since a higher recovery of the base cations was found with a 1 M HCl extraction (Fenn et al., 2018) compared to a 0.5M HCl extraction (Yamashita et al., 2014). In one test, we combined multiple molarities of HCl, resulting in an extraction sequence of 50 mL of 4 M HCl, followed by 50 mL of 2 M HCl and finally an extraction with 50 mL of 1 M HCl (Table 2). For both the KCl and HCl extractions, we varied the extraction volume, the extraction type and the extraction method (Table 2).

Extraction volumes used were 50, 100 and 150 mL, and extraction type was either single-column extraction or batch extraction. Using the single-column extraction type, the extractant was applied on the entire column, while in batch extraction the resin was divided into smaller samples. These subsamples of the resin were either fresh (i.e., solution drained resin) or dried at 28 °C to a constant weight (Table 2). Drying of the resin facilitates subsampling and the calculation of the deposition flux. The extraction method was either drip, in which the extractant was slowly dripped over the resin, or a shake–drip combination in which the resin was shaken in 50 mL of the extractant for 1 h and the remaining extractant was dripped over the resin. For shaking, the resin was put into a 50 mL centrifuge tube (Greiner Bio-One) with a screw cap. Then the resin was shaken using a speed of 120 movements per minute using a GLF 3015 platform shaker. After shaking, the resin was placed back into the original tube, and the extractant was allowed to drain from the resin and captured. The second 50 mL followed the drip procedure. The samples of the leachate of the macro- and micro-solution to load the columns, of the demineralized water to wash the loaded columns, and of the extraction of the elements from the columns were analyzed. We did not filter the samples as there was no visual contamination, and samples were handled under controlled laboratory conditions. Specifically, N–NH$_4^+$ and N–NO$_2$ + N–NO$_3^-$ concentrations were determined using a segmented flow analyzer (SFA type 4000, Skalar Analytical B.V., the Netherlands), while the content of Ca, Cu, Fe, Mg, Mn, Na, total P, S and Zn was analyzed using the ICP-AES (Thermo-Scientific iCAP 6500 DUO, USA).

## 2.3 Field tests

To evaluate the accuracy of the IER method to quantify bulk deposition and throughfall, a field study was carried out in the Netherlands (GPS 52.015745, 5.759924), in which we collected paired observations of bulk deposition and throughfall using water samples (referred to as the water method) and the IER method. The chosen field site consisted of a mature stand of European beech (*Fagus sylvatica*) which had been harvested at different intensities in February 2019, which resulted in four 0.25 ha plots within the same stand: an unharvested control ($\sim$ 0 % canopy openness), a high-thinning treatment ($\sim$ 25 % canopy openness), a shelterwood treatment ($\sim$ 75 % canopy openness) and a clear-cut treatment (100 % canopy openness) (Vos et al., 2023a, b). The different harvest intensities allowed the method to be tested for quantifying bulk deposition and throughfall, including the effect of organic substances on the performance of the IER method. The forest stand has a temperate maritime climate with a mean annual temperature of 10.4 °C and a mean annual rainfall of 805 mm (KNMI, 2022). An overview of the study site characteristics and placement of the paired samples is in Table 3.

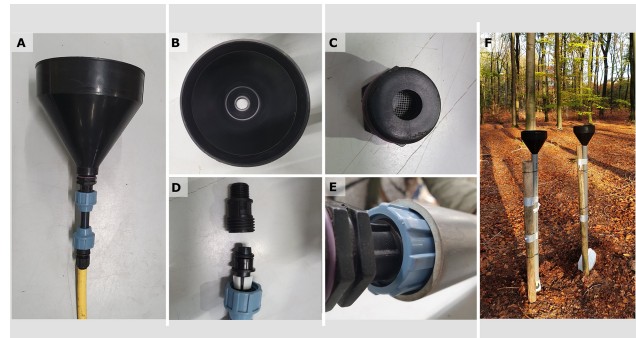

**Figure 2.** Construction of the deposition samplers. A: the connected sampler ready for use in the field. B: the used funnel with a collection surface of 288 cm$^2$. C: the wire coupling between the funnel and the resin column containing a mesh to prevent larger objects entering the resin column. D: overview of the resin column with the wire couplings. E: the resin column fitted tightly into the PVC tube, which allowed easy installation of the resin columns in the field. F: paired samplers in the field.

In each forest harvest treatment plot, 7 pairs of collectors were installed, resulting in 28 commonly used bulk and throughfall deposition collectors (collecting the precipitation next to and below the forest) and 28 IER deposition collectors. These 7 collectors per plot collectively had a collection surface > 2000 cm$^2$ above which the reliability of the measurement is significantly increased (Bleeker et al., 2003). The collectors consisted of a polyethylene funnel mounted to a resin column, which was filled with resin for the IER method but left empty for the water method, and a PVC hose connecting the resin column to a polypropylene water reservoir (Fig. 2A). The funnel had a surface of 288 cm$^2$ (including half the rim; Fig. 2B). Both the funnel and the resin column were chemically resistant and not susceptible to damage through UV light or low temperatures. Wire couplings, in which a mesh with the size of 0.51 mm was mounted, were used to connect the resin column to the funnel and to a hose tail (Figs. 2C, D). Prior to field installation, the funnel and the resin column including the wire couplings were cleaned from chemicals loosely bound to the surface by submerging into a 0.2 M HCl solution for 3 h, followed by 15 h immersion in demineralized water, which was continuously refreshed. Afterwards, the compartments were allowed to dry in a clean room and stored in clean plastic bags.

Field placement of the collectors was based on a digital elevation map of the canopy cover, assessed by drone-based photogrammetry (camera FC220). This digital elevation map was converted to a canopy cover map using "reclassify" in ArcMap (version 10.6.1) in which all data points above 10 m were assigned to be covered by canopy. Each plot was, thereafter, divided into an equal-sized, seven-block grid, and the locations of the collectors were determined in each of these blocks using random points, reflecting the canopy cover (%). Samplers were installed in the field using those

**Table 2.** Overview of the test for effective extraction of the ion exchanger. The KCl extraction was used for the extraction of $NH_4^+$ and $NO_3^-$, while the HCl extraction is used for the extraction of Ca, Mg, K, Fe, Mn, Zn, Cu, Na and S. The molarity of the extractant (M) was 2 for KCl and between 1–4 for the HCl extraction. In one case we used multiple molarities in one extraction, consisting of an extraction sequence of 50 mL of 4 M HCl, followed by 50 mL of 2 M HCl and finally an extraction with 50 mL of 1 M HCl. The single-column extraction included the entire loaded column (9.8 g of resin), while for batch extraction a subsample (avg 2.5 g dried resin) was used, which was either extracted fresh (i.e., solution-drained) or dried.

|  | Extraction solution | | Type | Resin | Samples | Method |
|---|---|---|---|---|---|---|
|  | M | mL |  |  |  |  |
| KCl | 2 | 50 | Single-column | Fresh | 3 | Drip |
|  | 2 | 100 | Single-column | Fresh | 3 | Drip |
|  | 2 | 50 | Batch | Fresh | 2 | Drip |
|  | 2 | 50 | Batch | Dried | 2 | Drip |
|  | 2 | 50 | Batch | Dried | 2 | Drip |
| HCl | 1 | 50 | Single-column | Fresh | 3 | Drip |
|  | 1 | 100 | Single-column | Fresh | 3 | Drip |
|  | 2 | 100 | Batch | Fresh | 2 | Drip |
|  | 2 | 100 | Batch | Dried | 2 | Drip |
|  | 2 | 100 | Batch | Fresh | 2 | Shake–drip |
|  | 2 | 100 | Batch | Dried | 2 | Shake–drip |
|  | 4 : 2 : 1 | 150 | Batch | Fresh | 2 | Drip |
|  | 4 : 2 : 1 | 150 | Batch | Dried | 2 | Drip |
|  | 2.5 | 100 | Batch | Dried | 2 | Shake–drip |
|  | 3 | 100 | Batch | Dried | 2 | Shake–drip |
|  | 3.5 | 100 | Batch | Dried | 2 | Shake–drip |

**Table 3.** Characterization of the study site and placement of the paired samplers in open gaps (bulk deposition) and underneath the forest canopy (throughfall). We statistically tested the behavior of the common water method and the IER method for throughfall and bulk deposition, regardless of the forest harvest intensity treatment in which these samplers were placed.

| Treatment | Canopy cover | Number of trees | Paired samplers | |
|---|---|---|---|---|
|  | % | Trees per hectare | Through-fall | Bulk deposition |
| Control | 94 | 245 | 6 | 1 |
| High-thinning | 72 | 180 | 5 | 2 |
| Shelterwood | 16 | 32 | 2 | 5 |
| Clear-cut | 0 | 0 | 0 | 7 |

random points on 6 November 2019, by placing the clean, connected sampler in the holder (PVC tube) and connecting the sample to the partly buried reservoir (Fig. 2F). The PVC tube was placed vertically so that the funnel, which was placed on top of the PVC tube, was aligned horizontally. The wire couplings of the resin column and the funnel fitted tightly into this PVC tube (Fig. 2E). Closed field blanks were installed simultaneously with the collectors, with one field blank in the clear-cut treatment (sun-exposed) and one field blank in the control (shade). Collectors and field blanks were

operational for 10 weeks. Funnel contamination (leaf litter and bird droppings) was recorded, and contaminated funnels were cleaned weekly. For the water method, the leachate was collected every week and sent to the laboratory. In the laboratory, the sample volume was recorded, and sample pH was measured, followed by sample filtration and measurement of the concentrations of Al, Ca, Cu, Fe, K, Mg, Mn, Na, P, S and Zn using the ICP-AES (Thermo-Scientific iCAP 6500 DUO, USA) and the concentrations of N–$NH_4$, N–$NO_3$ +N–$NO_2$, and inorganic carbon (IC) and total carbon (TC) using a segmented flow analyzer (SFA 4000, Skalar Analytical B.V., the Netherlands) within 24 h of sampling. The volume of the leachate of the IER collectors was collected monthly. The resin columns were collected on 14 January 2020 and dried together with lab blanks to a constant weight at 28 °C, and subsamples were taken for 2 M KCl extraction, followed by N–$NH_4$ and N–$NO_2$ + N–$NO_3$ concentration analysis using a segmented flow analyzer (SFA 4000, Skalar Analytical B.V., the Netherlands) and for 3.5 M HCl extraction followed by Ca, Cu, Fe, Mg, Mn, Na, P, S and Zn concentration analysis using the ICP-AES (Thermo-Scientific iCAP 6500 DUO, USA). There was no need to filter these samples as there were filters around the IER, and there were no visible undissolved particles. The 3.5 M HCl with a volume of 100 mL was chosen as an extractant because of reasonable extraction efficiency for P combined with reasonable extraction efficiencies for the other elements (except Zn). As a result of contamina-

tion by bird feces, only 18 out of the 28 paired collectors were used for the comparison. Uncontaminated paired collectors were evenly distributed between throughfall and bulk deposition.

## 2.4  Calculations and statistical analysis

The concentrations of the resin columns used in the laboratory and field test were corrected for the subsampling in the case of batch extraction, corrected for field and lab blanks and corrected for sample dilution prior to chemical analysis. To correct for subsampling, the concentration of the subsample was multiplied by CE1 the concentration of the entire column based on the weights of the subsample and the entire column respectively. Concentrations of the field and lab blanks were subtracted from the concentrations of the entire column to correct for field and lab contamination. For the bulk deposition samplers in the forest gaps, the sunlight-exposed field blank was used, and for the throughfall samplers underneath the forest canopy, the shadow field blank was used. Subsequently, the concentrations of the resin columns used in the field test were converted to the amounts per hectare for the entire measurement period. Thereafter, the deposition in $\mathrm{kg\,ha^{-1}}$ was calculated based on the funnel's surface. For the water method, the precipitation in $\mathrm{L\,ha^{-1}}$ was calculated based on the water volume per funnel (mL). Then the measured weekly concentrations were converted to $\mathrm{kg\,L^{-1}}$ and multiplied by the precipitation ($\mathrm{L\,ha^{-1}}$). Finally, for both methods, the samples were checked for bird droppings based on the P content, and samples with a P influx (in $\mathrm{kg\,ha^{-1}}$) larger than the mean $+2$ times the standard deviation were removed.

For the laboratory test, we calculated the adsorption capacity and the recovery efficiency. The adsorption capacity (i.e., percentage of total elemental influx adsorbed by the resin) was calculated as

$$\text{adsorption capacity} = \left(1 - \left(\frac{A_{\mathrm{out}}}{A_{\mathrm{in}}}\right)\right) \cdot 100,$$

in which $A_{\mathrm{in}}$ is the total amount of macro- and micronutrients in the solution (in µmol) applied to the resin, and $A_{\mathrm{out}}$ is the amount in the leachate (in µmol). The recovery efficiency (i.e., percentage of total elemental flux recovered from the resin) was calculated as

$$\text{recovery efficiency} = \frac{A_{\mathrm{ex}}}{A_{\mathrm{in}}} \cdot 100,$$

in which $A_{\mathrm{ex}}$ is the amount of macro- and micronutrients in the leachate of the applied extract (in µmol), which was poured over the loaded resin.

The adsorption capacity of the resin was evaluated using the Wilcoxon signed-rank test to test the hypothesis that adsorption is equal to 100 %. This test was only applied when the observed adsorption capacity was below 100 % to address

the issue of ties. Recovery efficiencies of lab extractions differing in molarity, resin pre-treatment and extraction type were tested using ANOVA type I error for unbalanced data following construction of a generalized least-squares model. Heterogeneity between groups was overcome using the varIdent weighting from the R package nlme (Pinheiro, 2017). Tukey's post hoc (honest significant difference) test was performed following ANOVA using the R package emmeans to test for differences between groups (Lenth et al., 2019). Goodness of fit between the original method and the IER method of the field test was tested using linear models using the lm function in the lme4 package (Bates et al., 2014). Outliers were removed from the linear models when Cook's distance was larger than $4/n$, where $n$ is the number of observations. The funnel position (throughfall or bulk deposition) was added as a random effect using lme models. This random effect was only retained if it improved model performance by $\Delta 2$, following Zuur et al. (2009), which proved to be true for none of the models.

## 3  Results

### 3.1  Adsorption capacity

The adsorption capacity of the resin (i.e., % of elemental flux bound to the resin) when loaded up to 70 % of the resin's exchange capacity was 100 % for all nutrients, with only Na and P being slightly lower (96 %–97 %) (Table 4). The adsorption capacity was not influenced by the flushing of the resin with demineralized water, indicating that the elements once adsorbed are not released through an excess of water like heavy precipitation.

Overloading the resin up to 150 % of the cation bed capacity resulted in decreased adsorption of $\mathrm{Na} > \mathrm{NH_4^+} > \mathrm{K}$ and a maximum loading of the cation bed of 115 %. Overload of the anion bed up to 160 % decreased the adsorption of $\mathrm{P} > \mathrm{NO_3^-}$. Increasing the elemental flux over the resin up to 230 % of the cation bed capacity and 240 % of the anion bed capacity resulted in lower adsorption of almost all elements except Ca and Zn (Table 4). Lab-controlled environmental conditions mimicking heat, drought and frost reduced the adsorption capacity of Na and P, and heat and drought slightly lowered the adsorption capacity of $\mathrm{NH_4^+}$. Elemental adsorption within the resin's exchange capacity was thus close to 100 % for all elements when the resin was used within its capacity, except for P, which was underestimated under the different simulated environmental conditions.

### 3.2  Recovery efficiency

The recovery efficiency of $\mathrm{NH_4^+}$ and $\mathrm{NO_3^-}$ under laboratory conditions (i.e., % of the elements that can be extracted from the resin) was generally high (mostly 90 %–100 %), with recovery depending on the molarity of the extraction (Table 5). The recovery efficiency of $\mathrm{NH_4^+}$ and $\mathrm{NO_3^-}$ did not differ be-

tween fresh and dry resin or between drip and shake–drip treatments using KCl extractions (ANOVA, $P$ value $< 0.05$, df: 20). We did not find differences between fresh or dry resin or between drip or shake–drip treatments using KCl extractions.

The average recovery efficiency following HCl extraction was high ($> 90\%$) for Ca, K, Na and Mn; slightly lower ($> 80\%$) for Mg, S, Cu and Fe; relatively low for P (40 %–91 %); and very low (6 %–25 %) for Zn (Table 5). Because extraction of Zn was unreliable, this element is not further included in average recovery numbers. The highest average recovery efficiencies were achieved with dried resin using either 2 M HCl extraction or 4 : 2 : 1 M HCl extraction. Specifically, the 2 M HCl methods yielded average recovery efficiencies of 94 % (drip) and 100 % (shake–drip), while the 4 : 2 : 1 M HCl method on dried resin achieved 90 % recovery efficiency. Recovery efficiency was significantly higher following an extraction on dried resin (avg recovery 88 %) compared to fresh resin (avg recovery 80 %), and recovery efficiency was slightly higher following shake–drip treatment (avg recovery 87 %) compared to drip-only treatment (avg recovery 84 %) (Table S3). We found an interaction effect between elements and pre-treatment, elements and molarity, and elements and extraction type, indicating that different elements responded differently to the different treatments. Overall, the highest average recoveries using HCl were found for the 2 M dry-weight shake–drip treatment, resulting in an average recovery of 100 %, whereas the lowest average recoveries (72 %) were found for the 1 M fresh-weight drip treatment (Table 5).

### 3.3 Performance under field conditions

There was a positive significant linear relationship between the deposition estimates of the water method and the IER method for all elements except for Ca, Zn and Fe (Table 6, Fig. 3). Absence of a relation for Ca, Zn and Fe was not related to the correction for contamination in blanks and for the lab recovery (Table 6).

The IER data corrected for contamination of blanks, and the lab recovery overall resulted in the highest $R^2$-adjusted values, resulting in a corrected goodness of fit up to 0.96 (K) and between 0.8 and 0.9 for $NH_4^+$, $NO_3^-$, S, Mg and Mn (Table 6). There was no difference between throughfall and bulk deposition between the IER method and the water method. For none of the elements did adding the position of the funnel (either throughfall or bulk deposition) increase the performance (expressed as the Akaike information criterion, AIC) of the statistical model. The IER method tended to have lower deposition estimates in the bulk deposition for Mg, Mn, Na and S, but overall, the IER method resulted in higher deposition estimates for $NH_4^+$, K, S, $NO_3^-$, Mg, Mn and Na compared to the water method (Fig. 3). For Fe, P and Cu, for which the water method yielded higher deposi-

**Table 4.** The mean ± SE of the adsorption capacity of the ion exchange resin following different tests ($n = 3$). Adsorption capacity is expressed as a percentage of the total elemental content that was captured by the resin. Tests included the loading of the resin with known concentrations within the resin's capacity (leachate); loading the resin beyond the resin's capacity (150 %–160 % and 230 %–240 % of the cation and anion bed, respectively); and resin pre-treatments including warmth, drought and frost. For adsorption capacities < 100 % we performed a Wilcoxon signed-rank test (WSRT) to see if these elements were different from a 100 % adsorption. The elements included in this test are marked in bold, and the number of samples used for the Wilcoxon test and the $P$ values are given.

| Test | Ca | Cu | Fe | K | Mg | Mn | Na | P | S | Zn | $NH_4^+$ | $NO_3^-$ | $n$ | WSRT $P$ |
|---|---|---|---|---|---|---|---|---|---|---|---|---|---|---|
| 70 % loading SE | 100±0.0 | 100±0.05 | 100±0.034 | 100±0.0 | 100±0.09 | 100±0.013 | **97±0.31** | **96±0.48** | 100±0.0 | 100±0.025 | 100±0.13 | **99±0.086** | 9 | 0.009 |
| 150 %–160 % loading SE | 100±0.1 | 100±0.067 | **85±3.7** | 100±0.0 | 100±0.15 | **66±2.9** | **59±2.0** | 99±0.31 | **74±4.4** | 99±0.11 | **93±3.2** | 99±0.0 | 24 | <0.001 |
| 230 %–240 % loading SE | **99±0.068** | **99±0.094** | **63±0.87** | **97±0.0** | **99±0.098** | **53±0.4** | **65±1.5** | **86±0.67** | **57±0.94** | **99±0.097** | **74±0.71** | 99±0.0 | 36 | <0.001 |
| Heat SE | 100±0.0 | 100±0.05 | 100±0.011 | 100±0.0 | 100±0.0 | **96±1.4** | **77±5.3** | 100±0.012 | **98±0.72** | 100±0.0 | | 100±0.012 | 9 | 0.004 |
| Drought SE | 100±0.2 | 100±0.05 | 100±0.011 | 100±0.0 | 100±0.045 | **96±1.1** | **79±3.9** | 100±0.0 | **99±1.1** | 98±0.69 | | 100±0.022 | 12 | 0.004 |
| Frost SE | 100±0.0 | 100±0.05 | 100±0.011 | 100±0.0 | 100±0.0 | **97±0.12** | **81±3.6** | 100±0.012 | 99±0.086 | 100±0.0 | | 100±0.012 | 9 | 0.009 |

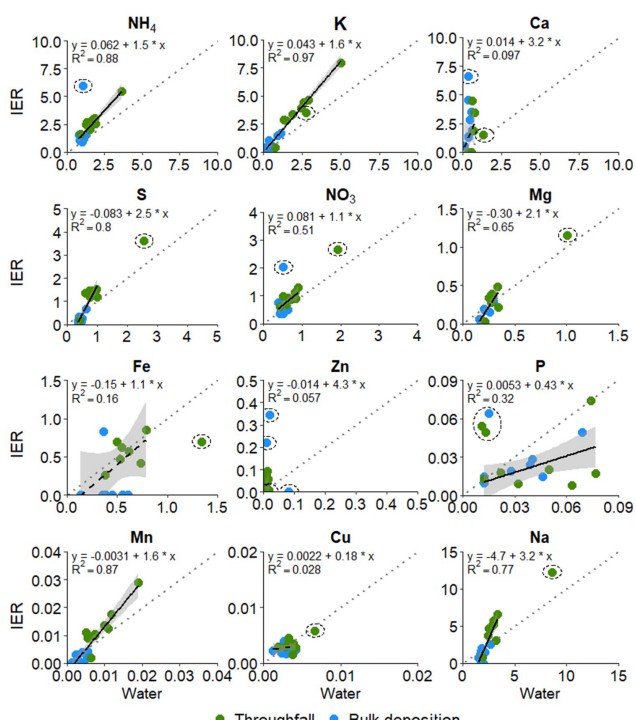

● Throughfall ● Bulk deposition

**Figure 3.** Relationship between the deposition estimates of the IER method ($\mathrm{kg\,ha^{-1}}$) and of the water method ($\mathrm{kg\,ha^{-1}}$) for the 10-week measurement period. Significant relationships are depicted with a solid black line and non-significant relationships with the dashed black line. The regression formula, the 95 % confidence intervals (grey) and the $R^2$ are shown. The standard errors and significance of the intercept and slope are given in Table 6. The 1 : 1 line is shown as the dotted grey line.

tion estimates, all values of the water method were below the detection limit (Table S4).

## 4 Discussion

### 4.1 Adsorption capacity

We aimed to test the capacity of IER as a method to quantify atmospheric deposition for a broad range of macro- and micro-elements, comparing results under laboratory and field conditions and in the latter case comparing bulk deposition and throughfall. First, the adsorption capacity of the IER, when loaded up to 70 % of its capacity as reported by the manufacturer, was generally high for all elements. High adsorption confirms earlier studies which found no elemental loss of $NO_3^-$, $NH_4^+$ and $SO_4$ (Simkin et al., 2004; Sheibley et al., 2012; Sheng et al., 2013) or only slight losses of $NH_4^+$ and $NO_3^-$ (Fang et al., 2011) and contradicts findings of low resin adsorption (Langlois et al., 2003). We show that IER is also able to adsorb above 99 % for a range of other elements, including the base cations and some micronutrients, and that the adsorbed elements are not released in response to an ex-

**Table 5.** Recovery efficiency of Ca, Cu, Fe, K, Mg, Mn, Na, P, S and Zn following HCl extractions and of $NH_4^+$ and $NO_3^-$ following KCl extractions. Recovery efficiencies are expressed as a % of the total elemental content poured over the resin. The efficiencies of the recovery were tested using extractions with different molarities (M), based on fresh (FW) or dried (DW) resin and based on drip or shake–drip treatments. The number of samples ($n$) for each extraction combination, the average arithmetic recovery per extraction combination (Avg) and the average arithmetic recovery for each element is given. Recovery percentages per element closest to 100 are indicated in bold. Differences in recovery efficiencies between elements are indicated with superscript letters a–f based on the average element recovery; test statistics are given in Table S3.

| M | Resin | Method | n | Ca | Cu | Fe | K | Mg | Mn | Na | P | S | Zn | Avg* | $NH_4^+$ | $NO_3^-$ |
|---|---|---|---|---|---|---|---|---|---|---|---|---|---|---|---|---|
| 1 | FW | Drip | 6 | 30±5.2 | 80±6.3 | 56±4.2 | 88±1.1 | 59±4.5 | 91±6.5 | 97±3.0 | 59±6.5 | 92±2.2 | 17±3.6 | 72 | **100±1.9** [TS1] | **98±5.0** |
| 2 | DW | Drip | 2 | 130±3.8 | 94±1.3 | 99±0.05 | 98±3.4 | 100±6.8 | **100±1.8** | 93±3.3 | 40±9.7 | 91±5.2 | 21±6.0 | 94 | 97±6.0 | 94±3.9 |
| 2 | DW | Shake-drip | 2 | 130±9.0 | **100±2.6** | 110±3.5 | **100±1.7** | **100±3.7** | 110±2.4 | **100±4.0** | 65±16 | **100±6.1** | 7.6±0.45 | 100 | 94±9.0 | 89±12 |
| 2 | FW | Drip | 2 | 85±2.1 | 81±0.45 | 88±1.5 | 86±5.0 | 84±6.2 | 91±1.7 | 82±5.8 | 43±9.2 | 96±3.5 | 25±10 | 82 | **80±6.1** [TS2] | 95±8.1 [TS3] |
| 2 | FW | Shake-drip | 2 | 82±11 | 80±8.6 | 87±6.2 | 87±8.6 | 81±13 | 88±9.0 | 84±8.4 | 69±19 | 93±5.1 | 5.4±0.45 | 83 | | |
| 2.5 | DW | Shake-drip | 2 | 86±1.2 | 78±0.65 | **100±0.60** | 100±3.1 | 81±4.5 | 81±0.20 | **100±1.3** | **91±0.70** | 78±11 | 6.1±0.4 | 86 | | |
| 3 | DW | Shake-drip | 2 | 82±5.6 | 71±6.3 | 67±8.9 | 77±6.9 | 77±9.1 | 77±7.8 | 99±2.8 | 84±1.5 | 70±1.2 | 6.1±0.6 | 81 | | |
| 3.5 | DW | Shake-drip | 2 | 99±0.0 | 89±0.45 | 80±1.9 | 88±8.1 | 93±1.9 | 96±2.4 | 84±5.2 | 83±4.1 | 83±6.9 | 11±2.6 | 88 | | |
| 4:2:1 | DW | Drip | 2 | **100±11** | 92±6.4 | 99±3.6 | 93±5.0 | 88±5.8 | 100±6.6 | 89±7.5 | 49±5.5 | 96±3.1 | 31±5.8 | 90 | | |
| 4:2:1 | FW | Drip | 2 | 86±0.75 | 85±0.55 | 90±2.9 | 100±23 | 80±1.9 | 92±0.60 | 82±1.6 | 49±0.60 | 97±3.6 | **44±10** | 85 | | |
| Arithmetic average | | | | 91[d] | 83[b,c] | 83[c] | 93[a,b] | 84[c] | 91[b] | 90[a] | 63[e] | 88[b] | 19[f] | | | |

\* Average without Zn as this element was unreliable using our extraction method.

**Table 6.** Regression coefficients (intercept and slope ± SE) and $R^2$ of models for the relation between the IER method and the commonly used method, including correction for blanks and lab recovery ($n = 18$). Differences between corrected, blank-corrected and recovery-corrected are related to the corrections for contaminations (blank-corrected) and for the recovery of the elements (recovery-corrected). The "corrected" rows show the data that are corrected for both the contamination and the recovery. The $R^2$-adj. values of the model with the best fit are highlighted in bold. For each model we calculated the mean absolute error (MAE) using this formula: MAE $= (1/n) * \Sigma |y_i\text{-}x_i|$. The MAE is the average absolute difference between the predicted and the observed IER data.

| | | $NH_4^+$ | $NO_3^-$ | S | P | K | Ca | Mg | Mn | Cu | Fe | Zn | Na |
|---|---|---|---|---|---|---|---|---|---|---|---|---|---|
| **Corrected** | Intercept | 0.062 ± | 0.081 ± | −0.83 ± | 0.0053 ± | 0.043 ± | 0.014 ± | −0.30 ± | −0.0031 ± | 0.0022 ± | −0.15 ± | −0.014 ± | −4.7 ± |
| | SE | 0.21 | 0.18 | 0.20*** | 0.0078 | 0.13 | 1.4 | 0.099** | 0.00085* | 0.00085* | 0.35 | 0.057 | 1.1*** |
| | Slope | 1.5 ± | 1.1 ± | 2.5 ± | 0.43 ± | 1.6 ± | 1.1 ± | 2.1 ± | 1.6 ± | 1.1 ± | 1.1 ± | 4.3 ± | 3.2 ± |
| | SE | 0.14*** | 0.30** | 0.31*** | 0.17* | 0.069*** | 1.6 | 0.40*** | 0.15*** | 0.27 | 0.69 | 5.0 | 0.45*** |
| | $R^2$ | **0.88** | **0.51** | **0.80** | **0.32** | 0.96 | 0.097 | **0.65** | **0.87** | 0.028 | **0.16** | **0.057** | **0.77** |
| | MAE | 0.31 | 0.16 | 0.19 | 0.0096 | 0.25 | 1.3 | 0.067 | 0.0021 | 0.00065 | 0.028 | 0.029 | 0.84 |
| **Blank cor.** | Intercept | 0.11 ± | −0.099 ± | −1.39 ± | 0.086 ± | 0.36 ± | −1.1 ± | −0.50 ± | −0.0049 ± | 0.0066 ± | −0.0044 ± | 0.014 ± | −8.0 ± |
| | SE | 0.18 | 0.13 | 0.50* | 0.035* | 0.30 | 3.5 | 0.27 | 0.0031 | 0.0014** | 0.018 | 0.016 | 2.6** |
| | Slope | 1.1 ± | 0.83 ± | 5.2 ± | −0.25 ± | 4.2 ± | 11 ± | 4.7 ± | 4.3 ± | 4.3 ± | 0.061 ± | −0.055 ± | 6.6 ± |
| | SE | 0.12*** | 0.23*** | 0.8*** | 0.81 | 0.17*** | 7.0 | 1.1*** | 0.43*** | 0.60 | 0.03 | 1.4 | 1.2*** |
| | $R^2$ | 0.82 | 0.46 | 0.72 | 0.00 | 0.97 | **0.14** | 0.50 | 0.85 | **0.30** | 0.15 | 0.00 | 0.66 |
| | MAE | 0.29 | 0.13 | 0.60 | 0.058 | 0.67 | 3.6 | 0.21 | 0.0063 | 0.0016 | 0.017 | 0.029 | 2.5 |
| **Recov. cor.** | Intercept | 0.15 ± | 0.15 ± | 0.070 ± | 0.0053 ± | 0.043 ± | 2.0 ± | 0.10 ± | 0.00087 ± | 0.0022 ± | 0.020 ± | 0.055 ± | −0.44 ± |
| | SE | 0.22 | 0.17 | 0.17 | 0.0078 | 0.13 | 1.8* | 0.10 | 0.0011 | 0.00085* | 0.0091* | 0.063 | 1.0 |
| | Slope | 1.5 ± | 1.1 ± | 1.6 ± | 0.43 ± | 1.6 ± | 3.5 ± | 1.3 ± | 1.3 ± | 0.18 ± | −0.0040 ± | 1.4 ± | 1.9 ± |
| | SE | 0.14*** | 0.26*** | 0.26*** | 0.17* | 0.069*** | 3.5 | 0.41** | 0.15*** | 0.27 | 0.018 | 5.5 | 0.43*** |
| | $R^2$ | **0.88** | **0.51** | 0.73 | **0.32** | **0.97** | 0.07 | 0.39 | 0.84 | 0.028 | 0.00 | 0.00 | 0.57 |
| | MAE | 0.32 | 0.15 | 0.16 | 0.0096 | 0.25 | 1.7 | 0.068 | 0.0018 | 0.00065 | 0.0074 | 0.03 | 0.69 |

* 0.05 <> 0.01. No star is not significant. ** 0.01 > < 0.001. *** < 0.001.

cess of water such as heavy precipitation. Therefore, IER can be loaded within the 70 % of its exchange capacity without risking lower elemental adsorption. However, slightly lower adsorption capacities were found for Na and P. These lower adsorption capacities are caused by the lower cation-exchanger affinity for $Na^+$ and lower anion-exchanger affinity for $HPO_4^{2-}$ (Skogley and Dobermann, 1996; Park et al., 2014). The lower adsorption capacities when using the resin within its capacity can lead to an underestimation of the total deposition of P by 4 %, although other studies report no lower adsorption capacities for P (Tahovská et al., 2016). Despite the possible underestimation of the total deposition, studies using IER report P deposition values within the natural ranges (Hoffman et al., 2019; Decina et al., 2018), indicating that the method usually also works well for P. The lower adsorption capacity of Na, however, can result in lower estimates of the deposition for multiple elements when Na is used as an tracer for canopy exchange processes (Staelens et al., 2008), and the use of Na as a tracer in IER deposition studies is thus questionable.

To further test the affinity of the resin for the studied elements, the resin was loaded to approximately 160 % and 240 % of its capacity. Based on the adsorption capacity beyond the resin's capacity, we found that the cation bed has an affinity of Ca = Fe > Cu = Mn = Zn > Mg > K > $NH_4^+$ > Na, which is in line with the previous reported resin affinity (Skogley and Dobermann, 1996). The anion bed has an affinity of S > $NO_3^-$ > P, which agrees with earlier studies (Skogley and Dobermann, 1996; Park et al., 2014). The resin's affinity and the adsorption capacity for different levels of loading beyond the resin's capacity are of importance for resin columns under suspicion of overloading. We did not find lower adsorption of Ca and Fe and only slightly lower adsorption of Cu, Mg, Mn and Zn, indicating that, when columns are slightly overloaded, these estimates are still reliable. When columns are loaded > 100 % of the capacity, the estimates for K, Na, P, S, $NH_4^+$ and $NO_3^-$ are not reliable. Therefore, in the case of suspicion of ion exchange overload, tests are recommended to check if stoichiometry between any element of Ca, Cu, Mg, Mn and Zn with K, Na, P, S, $NH_4^+$ and $NO_3^-$ falls within the stoichiometric range of natural deposition estimates. We strongly recommend collecting the resin columns prior to resin saturation as adsorption of Na and P can further decrease when saturating the resin up to 90 % or 100 %. The time period that the resin can stay in the field depends on the total atmospheric deposition and the volume of resin used. For remote areas with low deposition levels and low risk of sample contamination (e.g., by bird feces), the resin can stay for multiple months up to a year in the field as long as adequate resin volumes are used.

Heat, drought and frost treatments hardly influenced the absorption capacity of most elements but decreased the P adsorption capacity and, in the case of heat and drought, $NH_4^+$ and Zn (the latter only for drought) adsorption. These findings are in line with the adsorption behavior of some other IER types, where drying significantly reduced $NH_4^+$ adsorption, while frost–thaw cycles did not (Hart and Binkley, 1984; Kjonaas, 1999). However, in other work, extensive dry–wet cycles did not affect the adsorption of $PO_4$, $NO_3^-$ and $NH_4^+$ (Mamo et al., 2004), indicating that the effect of environmental conditions differs per resin type. Application of the IER method without an adsorption pre-test of the resin can therefore potentially underestimate $NH_4^+$ and P deposition when used in areas with temperatures above 40 °C and can potentially underestimate $NH_4^+$, P and Zn deposition in areas with longer drought periods. Despite the effect on some elements, weather circumstances generally seem to have little effect, indicating that the method is suitable under different climatic circumstances, like the boreal zone (Fenn et al., 2015), temperate zone (Hoffman et al., 2019; Fenn and Poth, 2004) and the tropics (Ibrahim et al., 2022; Kohler et al., 2012). The robustness of the method under different climatic circumstances implies that it can be used to compare deposition over large environmental gradients, which is essential to understanding regional and global deposition patterns.

## 4.2 Recovery efficiency

The recovery efficiency was tested based on differences in molarity, resin pre-treatment and extraction type. For this test, we used generally double or triple samples which can be considered a low sample size. However, because these tests were performed under controlled laboratory conditions, a small sample set can be justified. When using the IER method for field studies, we recommend testing the extraction solution for a larger number of samples to reduce the standard error (SE; Table 5). Nonetheless, we are confident that our conclusions are justified, given the controlled circumstances of the laboratory tests and the relatively low standard errors.

Recovery of $NH_4^+$ and $NO_3^-$ TS4 was highest following a 1 M KCl extraction based on controlled percolating of the extraction solution through the resin. Although the highest recovery following a 1 M KCl extraction has been reported before (Hart and Binkley, 1984), most studies indicate that 2 M KCl extractions will lead to higher recovery of both $NO_3^-$ and $NH_4^+$ (Kjonaas, 1999). However, the 2 M KCl recovery efficiencies of this study were comparable to other studies using 2 M KCl as an extractant (Tulloss and Cadenasso, 2015; Sheng et al., 2013; Fenn et al., 2002). The highest recoveries were obtained using dried resin and the combined shake–drip method.

Recovery efficiency following HCl extraction differed between elements and depended on the extraction itself. We choose HCl as an extractant as this extraction solution allows measurements of a broad range of elements using the ICP-AES, and this method was rarely tested. A limited number of studies used HCl as an IER extractant (Yamashita et al., 2014; Van Dam et al., 1987; Szillery et al., 2006; Dober-

mann et al., 1997), but only one study, testing only two elements, reported (high) elemental recoveries (Van Dam et al., 1987). Although $H^+$ has a relatively low affinity for the cation bed (Skogley and Dobermann, 1996), we expected that increasing molarities would increase recovery efficiency of both the cation and the anion bed. Surprisingly, recovery efficiency was highest using 2 M HCl and 4 : 2 : 1 M HCl, although highest recovery differed between elements (Table 5). Overall, we did find much higher recovery efficiencies for Ca and Mg using HCl extractions compared to KI and $H_2SO_4$ extractions (Wieder et al., 2016; Kohler et al., 2012), which can be related to a better extraction efficiency of HCl. Absence of higher recoveries using > 3 M HCl can be caused by differences in extraction time between treatments (Zarrabi et al., 2014) although the overall differences in recovery efficiencies between extractants were rather small.

  Recovery efficiency was higher when resin was dried prior to HCl extraction and when using the shake–drip extraction (Tables S5 and S7). The mechanism behind higher recovery efficiency following pre-extraction drying remains speculative but might be related to a better accessibility of the extract to reach micropores when the resin was dried. Previously, it was argued that pre-loading drying resulted in lower recovery efficiencies because of unavailable micropores due to swelling of the resin after rewetting (Kjonaas, 1999), but this unavailability of micropores was contradicted by Mamo et al. (2004), who found that dry–wet cycles significantly increased the desorption of elements from the resin. Occurrence of dry–wet cycles under field conditions can therefore interfere with the recovery efficiency of elements from the resin, which could possibly bias deposition estimates. This effect is, however, likely small as full drying resulted in only 8 % more efficient recoveries. The higher recovery following shake–drip treatment can result in longer contact time with the extractant (Zarrabi et al., 2014) while still avoiding the equilibrium reaction that occurs when using the shake treatment only. However, the present paper was not designed to test the effect of extraction time on the recovery efficiency; a complete test of this hypothesis will have to await future experimentation.

  Finally, the best extraction to use depends on the elements of interest. When studied elements are limited to the base cations, the 2 M HCl extraction provides good recovery efficiencies. However, studies including P and Zn should rather choose a HCl extract with a higher molarity or choose another extractant. Overall, recovery efficiencies of P and Zn were rather low, which may result from the low initial concentrations (Zarrabi et al., 2014). We did not test different extraction solutions as there are only limited options for extracting a broad range of macro- and micro-solutions. However, for P and Zn, different extraction solutions should be tested to increase the recovery efficiency. Furthermore, using the recovery efficiencies, we found only limited evidence of a release of background levels of elements from the resin. Indications of the release of background levels were present

for Ca (up to 130 % recovery) and Na (up to 110 % recovery). These indications were mainly present in the 2 M HCl dry-weight shake–drip extraction and could possibly be caused by lab contaminations. We did not find evidence for high background levels of $NO_3^-$ and $NH_4^+$, contrary to Langlois et al. (2003), who argued that the IER method was not suited for monitoring subtle patterns of $NO_3^-$ and $NH_4^+$ deposition. Together, our findings indicate that both KCl and HCl perform well as an extractant, except for P and Zn, for which new extraction methods should be tested.

## 4.3 Performance under field conditions

In general, deposition estimates based on the IER method were positively related to the deposition estimates of the water method; however, the IER method often resulted in higher deposition estimates. Exceptions were Fe and Ca, for which we did not find a relation between the deposition estimates of the IER method and the water method. This could indicate pollution related to elevated Ca and Fe leaching from the sample materials. For example, in the sun-exposed field blank we found high Fe pollution, causing the Fe deposition levels of all exposed collectors to be 0 (Fig. 3). For the collectors corrected for the shade-exposed field blank, we found good agreement between the deposition estimate of the IER method ($0.68\,\mathrm{kg\,ha^{-1}} \pm 0.12\,\mathrm{SE}$) and the water method ($0.66\,\mathrm{kg\,ha^{-1}} \pm 0.09\,\mathrm{SE}$), with the deposition estimates of both methods within the normal range of throughfall Fe deposition of the winter period (RIVM, 2015). For Zn we found much higher deposition values using the IER method compared to the water method in contrast to throughfall, which was much higher than bulk deposition estimates multiplied by the throughfall multiplication factor (Table S1). It could be that the presence of organic particles interfered with the recovery efficiency of Zn, possibly leading to an overestimation of the Zn throughfall.

  The higher deposition estimates of the IER method compared to the water method for $NH_4^+$ and $NO_3^-$ can be caused by absence of biochemical reactions, which causes losses of these elements in the original samplers (Kohler et al., 2012; Fenn and Poth, 2004). Higher deposition estimates using the IER method can also be related to the low concentration of elements in the water method, which were often below the detection limit (Table S4). Overall, slightly higher deposition values using IER columns were reported before (Fenn and Poth, 2004; Simkin et al., 2004; Kohler et al., 2012). Because of the absence of biochemical reactions and higher reliability of the lab measurements for IER samples, the IER method is likely more reliable to quantify both bulk deposition and throughfall compared to the water method, and the generally higher deposition estimates are likely a better representation of the actual atmospheric deposition.

  The lower deposition estimates of P can be caused by a better adsorption of inorganic P compared to organic P to the resin (Zarrabi et al., 2014), which potentially reduces the

recovery efficiency of P under field conditions compared to lab conditions. However, additional field tests are necessary for P to compare the difference between field and laboratory adsorption and recovery efficiencies. We attempted to extract $PO_4^{3-}$ from the resin using 2 M KCl as an extractant but achieved only an 8 % recovery ($n = 4$, data not shown). To determine if the lower deposition estimates of P are due to better adsorption of inorganic P compared to organic P, further efforts are needed to successfully extract $PO_4$ from the resin. For other elements, the comparison of the IER method and the water method did not give evidence of lower adsorption or recovery efficiencies under field conditions. Absence of this effect might, however, be related to the winter period in which the field measurements took place as, for example, pollen was hypothesized to reduce recovery of $NH_4^+$, $NO_3^-$ and $SO_4$ from the IER (Brumbaugh et al., 2016). Lower field recovery might, therefore, beside the resin type and the extraction method, be related to the amount of organic particles like pollen, which was not included in this study.

## 5 Conclusions

We tested the suitability of the IER method for quantifying bulk deposition and throughfall of macro- and micronutrients by assessing adsorption capacities and recovery efficiencies under controlled laboratory conditions, followed by an evaluation of the performance of the method under field conditions.

Results showed the following:

1. The adsorption capacity of the resin under controlled laboratory conditions was close to 100 % for all nutrients.

2. Extraction using KCl is effective for nitrogen ($NH_4^+$ and $NO_3^-$) with general high recoveries (mostly 90 %– 100 %) depending on the molarity of the extraction TS5, while extraction using HCl is effective for Ca, K, Na, Mn, Mg, S, Cu and Fe but not for P and Zn, for which testing using other extraction methods or extraction solutions is recommended.

3. Drying the resin prior to extraction and using a shake–drip extraction method increased the recovery efficiencies.

4. The IER method is useful under a broad range of environmental conditions, since heat (40 °C), drought and frost (−15 °C TS6) hardly affected the adsorption of nutrients except for P, which was reduced up to 25 %.

5. The IER method performed well under field conditions, resulting in similar but consistent higher deposition estimates compared to the water method.

Our results even imply a higher reliability of the IER method than the water method under certain circumstances since uncertainties related to biological reactions and the detection limit for lab measurements could be removed. However, possible contamination of the IER collectors due to factors such as bird feces or other animal disturbances is a point of concern, as long field exposure increases the risk of contamination. It is therefore recommended to increase the number of samplers when using the IER method. We conclude that IER is a powerful tool for monitoring the element input by bulk deposition and throughfall for a broad range of elements, across a broad range of environmental conditions.

*Data availability.* No data have been published yet. Data will be made available upon request.

*Supplement.* The supplement related to this article is available online at: https://doi.org/10.5194/amt-17-1-2024-supplement.

*Author contributions.* MAEV conceived the ideas and designed the methodology. Prior to testing the methods, the ins and outs were discussed with WdV and FJS. MAEV tested the ion exchange resin method both in the laboratory and in the field and prepared the samples for chemical analysis, which was ultimately performed by the laboratory staff. Statistical analysis was performed by MAEV. MAEV led the writing of the manuscript. All authors contributed critically to the drafts and gave final approval for publication.

*Competing interests.* The contact author has declared that none of the authors has any competing interests.

*Acknowledgements.* This research is part of the Nutrient Balance project and was funded by the Dutch Research Council (NWO, grant no. ALWGS.2017.004). We thank Henk van Roekel and the staff of the CBLB laboratory and Tupola, Wageningen University, for their valuable assistance in developing and testing this method. We also express our gratitude to our partners the State Forestry Service, the Forest Owner Association, Het Loo royal estate, Staro Nature and Countryside, Borgman Forestry Consultants, De Hoge Veluwe National Park and Blom Ecology for their financial support and permission to conduct research in their forests

*Financial support.* This research has been supported by the Nederlandse Organisatie voor Wetenschappelijk Onderzoek (grant no. ALWGS.2017.004).

*Review statement.* This paper was edited by Mingjin Tang and reviewed by three anonymous referees.

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

**Remarks from the language copy-editor**

CE1    Please note that "by" is the correct preposition to be used with "multiplied". Do you perhaps mean a different verb (e.g. "scaled" or similar) to describe what is being done with these concentrations?

**Remarks from the typesetter**

TS1    Please provide a statement for the editor for this change. Thank you.

TS2    This change of values needs to be approved by the handling editor. Please write a statement to explain why this change needs to be made.

TS3    This change of values needs to be approved by the handling editor. Please write a statement to explain why this change needs to be made.

TS4    Please note that even if the reviewers commented on this, like the values in the table, we still need approval from the editor to remove chunks of text like this. The typesetter and copy-editor can unfortunately not judge whether this has been approved. Many thanks in advance for your cooperation and understanding. Please prepare a short statement which we can forward to the editor.

TS5    Please write a statement for the handling editor to explain why this needs to be removed.

TS6    Please write a statement for the handling editor to explain why this has to be changed.

TS7    Please check DOI number, it leads to a different reference.