# Peer review of "Testing Ion Exchange Resin for quantifying bulk and throughfall deposition of macro and micro-elements on forests"

_Atmospheric Measurement Techniques, 2024_

## Referee Comment (RC2)

**Testing Ion Exchange Resin for quantifying bulk and throughfall deposition of macro and micro-elements on forests**

Vos, Marleen A.E, et al.

Overview

This work provides insights into using ion exchange resins for precipitation sampling, both in a controlled laboratory setting and field studies. This is interesting and relevant work, as IER are under used and could provide a cheaper more robust alternative for precipitation sampling in a wide variety of environments. The introduction emphasizes the importance of this work well.

The laboratory studies highlight the feasibility of this approach for a wide variety of analytes under several conditions that mimic the environment, as well as determine an efficient method for extracting analytes from the resin. The field studies show the application of this technique in practice. However, the field studies do have a limited sample size, which could impact their reliability.

I recommend this manuscript for publication in Atmospheric Measurement Techniques, following major revisions. Overall, the statistical analysis of the data is unclear and requires clarity so that readers can be confident in the use of IER for sampling. While the statistical analysis is the major point of concern, please refer to my specific comments below.

Specific Comments

Lines 138-142: I think this info regarding stock solutions and concentrations tested for each nutrient would be easier to read as table. Furthermore, ensure that all chemicals have numbers subscripted.

2.2 Laboratory Tests: were blanks (water with no nutrients in solution) analyzed for both the adsorption capacity and recovery efficiency test? These should be described somewhere within this section.

Line 237: Going back to my previous comment, you mention field and lab blanks here. Be sure to include information on how both were prepared in section 2.2.

Line 265: I am assuming you highlighting data in the "leachate" row of Fig.3, however I think it would be useful to explicitly direct the reader to that portion of the figure.

Figure 3: Do these represent average values across several trails? Finding a way to include standard deviations for this data, if so. This might help support your claim that certain nutrients had decreased capacity when loading was increased. Were t-tests done (or any statistical analysis) to support that these values are indeed different from each other?

Lines 289-290: This sentence is really confusing and I was having a difficult time connecting it to the data presented in Table 3.

Line 292: What does 4-2-1M indicate? This needs more description.

Table 3: Is there a way to incorporate the statistics presented in Table S3 into this main table?

Table 3: What does the column labeled "Mol" represent? I'm seeing here the dashed values again (ex: 1-2). This needs some clarity. In addition, is the Avg column necessary? This accounts for some poor recoveries for some species and artificially makes the method look reasonable. I think this will allow readers to justify using a less than optimal extraction technique.

Figure 4: Do the percentages indicate canopy cover? Please provide that detail in the figure caption.

Table 5: Either ensure that the variations in the intercept and slope are on the same line or create a separate row title for these values. It's challenging to read.

Line 324: Why was an ANOVA done of the comparison of IER method to water-method if there were only two categories? If I'm missing additional categories, then this needs emphasized.

Line 325: What were the treatments analyzed using the Tukey's test? The lines surrounding this sentence could use additional clarification so the reader understands what is being compared.

Line 326: I'm not sure how Figure S1 is displaying statistical data.

Line 326: Sometimes the authors refer to canopy openness in terms of percentages and other times using words like "clear cut". Using consistent terminology would be useful if these are supposed to represent the same samples.

Line 328: Is these supposed to refer the reader to Figure S4? If not, I'm missing how Fig. 4 connects to your statement.

Line 335: How was the loading capacity for the resin determined? If obtained from the supplier this might be pertinent information to include in the methods section (either directly in the text or as a supplemental table).

Line 339: How did you determine this 70% if you didn't test below 100%? This statement is confusing.

Section 4.1 Adsorption capacity: In general, how can there still be 100% adsorption if the resin is loaded above its capacity? Is this trying to emphasize that the determined capacity is an underestimation?

Line 390: Is the highest recovery for each element bolded in Table 3. If may be useful to indicate that and provide an explanation for why values over 100 were not considered.

Line 395: I am not sure how this statement related to the data provided in Table S3. Provide some clarity here.

---

## Author Comment (AC1)

**Reviewer 2**

Overview

This work provides insights into using ion exchange resins for precipitation sampling, both in a controlled laboratory setting and field studies. This is interesting and relevant work, as IER are under used and could provide a cheaper more robust alternative for precipitation sampling in a wide variety of environments. The introduction emphasizes the importance of this work well. The laboratory studies highlight the feasibility of this approach for a wide variety of analytes under several conditions that mimic the environment, as well as determine an efficient method for extracting analytes from the resin. The field studies show the application of this technique in practice. However, the field studies do have a limited sample size, which could impact their reliability. I recommend this manuscript for publication in Atmospheric Measurement Techniques, following major revisions. Overall, the statistical analysis of the data is unclear and requires clarity so that readers can be confident in the use of IER for sampling. While the statistical analysis is the major point of concern, please refer to my specific comments below.

➢ We thank the reviewer for the positive evaluation and suggestions for improving the provided statistics: we modified the applied statistics according comments of this reviewer (see below), and others. See for details our replies below.

Specific Comments

Lines 138-142: I think this info regarding stock solutions and concentrations tested for each nutrient would be easier to read as table. Furthermore, ensure that all chemicals have numbers subscripted.

➢ The subscripts have been adjusted to meet the correct standards. The molecular formula of $Cu(NO_3)2.3H_2O$ has been changed to $Cu(NO_3)_2 \cdot 3H_2O$, and the molecular formula of $Zn(NO_3)2.6H_2O$ has been changed to $Zn(NO_3)_2 \cdot 6H_2O$. Given the density of information in these lines, we agree with the reviewer that presenting the content in a table will enhance readability. The table replacing lines 138-142 is shown below.

**Table 1: The throughfall flux used to test the adsorption capacity and recovery efficiency of the ion exchange resin. We used stock solutions with known molarity to make the macro and the micro solution used to drip through the resin. The total volume of the used stock solution (in ml L$^{-1}$) and the concentration in umol per element are given.**

| Stock solution | | Type | Total | Ca | Cu | Cl | Fe | K | Mg | Mn | Na | PO$_4$ | SO$_4$ | Zn | NH$_4$ | NO$_3$ |
|---|---|---|---|---|---|---|---|---|---|---|---|---|---|---|---|---|
| Code | Mol | | *mL L$^{-1}$* | | | | | | | | *umol* | | | | | |
| Na$_2$SO$_4$ | 0.5 | Macro | 0.90 | | | | | | | | 450 | | 450 | | | |
| NaCl | 1 | Macro | 1.40 | | | 1400 | | | | | 1400 | | | | | |
| KNO$_3$ | 1 | Macro | 0.18 | | | | | 180 | | | | | | | | 180 |
| KH$_2$PO$_4$ | 1 | Macro | 0.02 | | | | | 20 | | | | 20 | | | | |
| NH$_4$NO$_3$ | 1 | Macro | 1.82 | | | | | | | | | | | | 1820 | 1820 |

| | | | | | | | | | | | | | | | | |
|---|---|---|---|---|---|---|---|---|---|---|---|---|---|---|---|---|
| NH₄Cl | 1 | Macro | 2.18 | | | 2180 | | | | | | | | | 2180 | |
| MgSO₄ | 1 | Macro | 0.3 | | | | | | 300 | | | | 300 | | | |
| CaCl₂ | 0.5 | Macro | 0.8 | 400 | | 400 | | | | | | | | | | |
| FeCl₂ | 0.1 | Micro | 6.0 | | | 600 | 600 | | | | | | | | | |
| Cu(NO₃)₂·3H₂O | 0.275 | Micro | 0.036 | | 9.9 | | | | | | | | | | | 9.9 |
| Zn(NO₃)₂·6H₂O | 0.267 | Micro | 0.075 | | | | | | | | | | | 20 | | 20 |
| MnSO₄·H₂O | 0.01 | Micro | 15.0 | | | | | | | 150 | | | 150 | | | |
| | | | **total** | 400 | 9.9 | 4580 | 600 | 200 | 300 | 150 | 1850 | 20 | 900 | 20 | 4000 | 2030 |

2.2 Laboratory Tests: were blanks (water with no nutrients in solution) analyzed for both the adsorption capacity and recovery efficiency test? These should be described somewhere within this section.

➢ Yes, we had three different types of blanks including a blank for adsorption capacity, a blank for the recovery efficiency test in the laboratory and a blank for the possible field contaminations. Further details on these blanks are given below:

- Blank for adsorption capacity: during the laboratory tests, we tested the quality of the demineralized water that was used to flush the resin. This water was used to flush 500 grams of resin which we did prior to filling the resin tubes with this resin (lines 112-114). We took the sample from the first two liters which we used to flush this resin, in this sample there was no contamination as elemental concentrations in this sample could not be detected or was more than 10 times lower than the detection limit. As there could not be any element be detected in this blank sample, we are confident that the elements in the leachate of the adsorption test (see row "70% loading" in the revised table 4) are a result of the lower affinity of the resin to adsorb these elements (Na, P and NH₄). We added in the manuscript (lines 163 in the revised version) the following sentence to indicate the existence of this blank: "Furthermore, demineralized water used to clean the resin was taken as a blank sample for the adsorption test".

- Blanks for recovery efficiency test in the laboratory: these blanks consisted of pure resin that was treated exactly the same way as the resin used for the extraction tests. We included these blanks to remove noise due to sample contamination in the laboratory which might have been caused by the drying process, the extraction of the resin itself or by the measurement of the chemical content of the sample. We added in the manuscript (lines 165-167 in the revised version) the following sentence to indicate the existence of these blanks: "Besides these 36 loaded columns, 2 blanks were included to distinguish between the recovery efficiency of the loaded solution and background contamination out of the resin or contamination caused sample contamination in the laboratory".

- Field blanks: these blanks were placed in the field both in sun exposed place (representative for the bulk deposition samples) and in a shadowed place (representative for the throughfall samples). These blanks stayed in the field for the same time as the field-test samples and were extracted with the extractant. The field blanks were subject to the lab protocol meaning that these field blanks were extracted with 2M KCl

and with 3.5M HCl following the protocol developed in the lab. These blanks are mentioned in the manuscript both for the field placement (lines 291-292 in the revised version) as well as the laboratory extraction (lines 302-306 of the revised version).

Line 237: Going back to my previous comment, you mention field and lab blanks here. Be sure to include information on how both were prepared in section 2.2.

➢ The general description of the preparation of the columns is in section 2.1. Blank columns were created using the same protocol. To explicitly include this, we added the phrase "(including the blanks)" in line 112 of the revised manuscript. Furthermore, in section 2.2., we now explicitly describe the sampling of the leachate of the washed resin as a blank for the adsorption test (line 182 of the revised manuscript) and the sampling of the blanks used for the recovery efficiency tests (lines 196-198 of the revised manuscript). The field blanks were already described in the manuscript in section 2.3.

Line 265: I am assuming you highlighting data in the "leachate" row of Fig.3, however I think it would be useful to explicitly direct the reader to that portion of the figure.

➢ Your assumption is correct. For clarification, we changed the name "leachate" to 70% loading, and we added the phrase "when loaded up to 70% of the resins exchange capacity" in line 265. We now direct the reader to the relevant part of that table.

Figure 3: Do these represent average values across several trails? Finding a way to include standard deviations for this data, if so. This might help support your claim that certain nutrients had decreased capacity when loading was increased. Were t-tests done (or any statistical analysis) to support that these values are indeed different from each other?

➢ Yes, the data represent average values. The number of samples for each test is provided in lines 143-155 of the manuscript. To address the confusion around the figure, we replaced it with a table that includes information on the standard error of the mean. For all the tests presented in Table 4, we used a sample size of 3 samples per test. We have now specifically included the sample size in the table header.

➢ In response to your feedback, we conducted a generalized test per treatment group, as individual element-specific tests were impractical due to consistently low sample sizes and standard errors, resulting in essentially constant data that lacks statistical evaluability. For elements where mean adsorption was less than 100%, we used the Wilcoxon signed rank test to assess the hypothesis of adsorption equality to 100%. Excluding the values equal to 100% was necessary to deal with ties.

Lines 289-290: This sentence is really confusing and I was having a difficult time connecting it to the data presented in Table 3.

➢ We simplified the sentence from "Recovery efficiency of Ca, Cu, Fe, K, Mg, Mn, Na, P, S and Zn following HCl extraction was high (>90%) for Ca, K, Na and Mn, slightly lower (>80%) for Mg, S, Cu and Fe, relatively low for P (40-91%) and very low (6-25%) for Zn (Table 4)" to "The average recovery efficiency following

HCl extraction was high (>90%) for Ca, K, Na and Mn, slightly lower (>80%) for Mg, S, Cu and Fe, relatively low for P (40-91%) and very low (6-25%) for Zn (Table 4)". We also rewrote the next sentences for readability.

Line 292: What does 4-2-1M indicate? This needs more description.

➢ We also tested if varying molarities from the extraction fluid would result in a more effective extraction of the resin. The 4-2-1 M means that we first extracted the resin with an 4M HCl extractant of 50mL, followed by a extraction with a 3M HCl extractant of 50mL followed by an extraction with 1M HCl of 50mL. We added the description in the manuscript in lines 201-203 of the revised version: "In one test, we combined multiple molarities of HCl, resulting in an extraction sequence of 50 mL of 4M HCl, followed by 50 mL of 2M HCl, and finally an extraction with 50 mL of 1M HCl (Table 2)". Furthermore, we added an extra explanation to the header of table 2.

Table 3: Is there a way to incorporate the statistics presented in Table S3 into this main table?

➢ We chose to add the statistics shown in Table S3 to the newly added Tables S6-S8 for better readability. The post-hoc test of each interaction results in different groups, making it challenging to include the complete statistics correctly and interpretably in Table 3 (now Table 5 in the revised manuscript). Even including the general ANOVA results (presented in Table S3) in Table 3 (Table 5 in the revised manuscript) is challenging and, in our opinion, would not enhance readability.

Table 3: What does the column labeled "Mol" represent? I'm seeing here the dashed values again (ex: 1-2). This needs some clarity. In addition, is the Avg column necessary? This accounts for some poor recoveries for some species and artificially makes the method look reasonable. I think this will allow readers to justify using a less than optimal extraction technique.

➢ Mol was used as an abbreviation of the molarities. We added this abbreviation to the table header.

➢ In the revised version of the manuscript, we choose to remove the test showing 1-2M extraction (which actually meant that we first extracted the sample with 50 mL 1M HCl followed by 50 mL of 2M HCl) because we lacked replication. Fur the 4-2-1 M test we added the descriptions as explained in the response to your previous comment.

➢ The columns providing the arithmetic averages are indeed not necessary, but we believe they give a good overview of the overall performance of the different extractions, guiding the reader toward the extraction method that might be best for their own research. Furthermore, the average recoveries of the elements across different tests indicate how easily each element can be extracted from the ion exchange resin. For example, if researchers are interested in using the IER method for Zn, this information directly warns them that Zn extraction from the resin is problematic.

Figure 4: Do the percentages indicate canopy cover? Please provide that detail in the figure caption.

➢ Yes, these percentages indeed reflected the canopy cover. However, in response to the comment, we decided to move away from the artificial groups based on canopy cover and instead used the funnel positions in the

statistics and in the figure. In the new Figure 4 of the revised manuscript, we refer to throughfall and bulk deposition, avoiding the confusion caused by the canopy cover percentages.

Table 5: Either ensure that the variations in the intercept and slope are on the same line or create a separate row title for these values. It's challenging to read.

➢ We added a separate row title, namely s.e., in front of every row containing the standard errors of the mean.

Line 324: Why was an ANOVA done of the comparison of IER method to water-method if there were only two categories? If I'm missing additional categories, then this needs emphasized.

➢ Originally, we included the treatment plot where the samples were placed as a categorical variable. This variable included the treatments control (high canopy cover) till clearcut (no canopy cover). For details on these categories see table 2. However, upon reflection, we realized our main focus was comparing the water-based deposition method with the ion exchange resin method in forest gaps (bulk deposition) and beneath the canopy (throughfall). In line with this, we revised the field study statistics to exclude the treatments (control, high-thinning, shelterwood, and clearcut), focusing instead on the funnel's position (throughfall or bulk deposition). We ran linear models for all elements with the funnel position as a random structure. These models were constructed as $y_{ij}=\beta 0+\beta 1 x_{ij}+u_j+\epsilon_{ij}$ , where $y$ is the result using the IER method, $x$ is the result using the original method, $u$ represents the random structure, and $\epsilon$ the residuals. The random structure was included only if it improved model performance by $\Delta 2$, following Zuur et al. (2009). None of the models showed improved performance with the funnel position as a random variable. The results are visualized in the revised Figure 4. Line 324 and other lines were we referred to the canopy cover, or canopy openness plots were revisited and rewritten.

Line 325: What were the treatments analyzed using the Tukey's test? The lines surrounding this sentence could use additional clarification so the reader understands what is being compared.

➢ In this version, we excluded the canopy openness treatments because they were not our primary focus. Line 325, which addressed the canopy openness treatments in the original manuscript, has been removed. Instead, results are added comparing the relationship between the IER-method and the original water method for bulk deposition and throughfall samples.

Line 326: I'm not sure how Figure S1 is displaying statistical data.

Figure S1 previously presented the data without highlighting between-group significances, which was an oversight. Following the revised statistical approach mentioned in our previous response, we updated Figure S1 to show the differences between bulk deposition and throughfall. As incorporating these differences did not enhance the linear models for the relationship between the IER samples and the common water samples (as shown in Figure 4 and Table 6 of the revised version), there were no statistical differences to report.

Line 326: Sometimes the authors refer to canopy openness in terms of percentages and other times using words like "clear cut". Using consistent terminology would be useful if these are supposed to represent the same samples.

➢ In the revised version, we excluded the canopy openness treatments because they were not our primary focus. Line 326, which addressed the canopy openness treatments like a clearcut treatment has been removed. Instead, a discussion is added comparing the relationship between the IER-method and the original water method for bulk deposition and throughfall samples.

Line 328: Is these supposed to refer the reader to Figure S4? If not, I'm missing how Fig. 4 connects to your statement.

➢ This statement has been adapted to compare the differences between bulk deposition and throughfall which is now clearly shown in the revised version of figure 4.

Line 335: How was the loading capacity for the resin determined? If obtained from the supplier this might be pertinent information to include in the methods section (either directly in the text or as a supplemental table).

➢ Yes, the loading (exchange) capacity of the resin was obtained from the manufacturer. We included this information in the method, in section 2.2 (lines 178-179 of the revised manuscript): "The exchange capacity of the resin as reported by the manufacturer was $\geq 0.6$ mol L$^{-1}$ for the anion bed and $\geq 0.7$ mol L$^{-1}$ for the cation bed".

Line 339: How did you determine this 70% if you didn't test below 100%? This statement is confusing.

➢ We loaded the resin up to 70% of the exchange capacity as reported by the manufacturer (see response to the previous comment). To specify this in this line, we added the text to: "First, the adsorption capacity of the IER, when loaded up to 70% of its capacity as reported by the manufacturer, was generally high".

➢ We did not make a statement that we didn't test the adsorption capacity of the IER below 100%. We tested this by loading the resin to 70% of it's capacity. However, to avoid this confusion, we changed original figure 3 into a table and replaced the term 'leachate' to '70% loading'.

Section 4.1 Adsorption capacity: In general, how can there still be 100% adsorption if the resin is loaded above its capacity? Is this trying to emphasize that the determined capacity is an underestimation?

➢ Overloading the cation and/or anion exchange capacity did not result in 100% adsorption capacity for all elements. We observed that only certain elements, which strongly bind to the resin, achieved (nearly) 100% adsorption capacity when the resin was overloaded beyond its exchange capacity. This suggests that the resin has a high affinity for these specific elements. In contrast, other elements, which have lower affinity for the resin, showed lower adsorption capacity under similar conditions. With other words, when the resin is loaded beyond its capacity, it releases loosely attached elements (like K, Na, NH4 and NO3) while still adsorbing more strongly binding elements (like Ca and Fe).

➢ Furthermore, the capacity reported by the manufacturer was indeed a slight underestimation of the actual exchange capacity. However, we decided not to include these details in the manuscript as they do not add significant value to the main message of this work.

Line 390: Is the highest recovery for each element bolded in Table 3. If may be useful to indicate that and provide an explanation for why values over 100 were not considered.

> ➢ Not the highest but the value closest to 100%. The purpose of the recovery efficiency tests was to find the extraction that results in the recovery efficiency that was closest to 100%. Ideally this would have been the same as the highest recovery efficiency, however, we clearly had a Ca contamination in some of the samples resulting in recovery efficiencies above 100% which should have been impossible. This was explained in the header of table 3 (table 5 in the revised version): "Recovery percentages per element closest to 100 are indicated in bold" .

Line 395: I am not sure how this statement related to the data provided in Table S3. Provide some clarity here.

> ➢ We changed the reference to the tables S5 and S7. These supplementary tables are added in the revised version of the manuscript and contain the average data and the results of the Tukey's post-hoc tests.

**Literature**

Zuur, A., Ieno, E.N., Walker, N., Saveliev, A.A., Smith, G.M., 2009. Mixed effects models and extensions in ecology with R. Springer Science & Business Media.

---

## Author Comment (AC2)

**Reviewer 3**

I would like to congratulate the authors for a very thorough work. A multitude of variables have been taken into account and the experimental and field design are excellent. It is a very necessary work from the point of view of forest monitoring, since the use of methodologies based on ion exchange resins is not widespread despite being used for decades already, and it could be because of the scarcity of methodological approaches such as the one here it is presented.

> ➢ We would like to thank the reviewer for the kind words. We fully agree with the reviewer's opinion that the ion exchange resin method has the potential to become a widely adopted technique if thoroughly tested under various conditions, such as different climatic regions, sampler types, field durations, and more. When we initiated the testing of this method, we recognized the scarcity of studies on how to test the resin and assess its reliability. This gap in research motivated us to write this manuscript, aiming to address these issues comprehensively.

I would like to recommend the publication of this manuscript, but not before suggesting some minor changes and raising some questions:

Line 108: First, the resin columns were cleaned using 0.2M HCl and demineralized water - Please add some few words to fully explain the cleaning method. Demineralized water was used for cleaning or for rinsing (as stated in the next sentence)?

> ➢ We used 0.2M HCl to clean the empty resin columns, removing weakly attached chemicals from the materials. These chemicals have the potential to transfer from the column walls to the resin beads during subsequent use, introducing contaminants into both laboratory and field tests. Since we utilized newly constructed resin columns, there was no need to address residuals from previous measurement sets. However, we recommend washing the columns again post-use. We added the reason why we cleaned the resin columns to the manuscript.

> ➢ After cleaning the empty resin columns with the 0.2M HCl solution, we ensured the removal of any remaining HCl. The reason why we cleaned the columns was added to the revised manuscript. Residuals of this fluid can be adsorbed by the resin, thus reducing its capacity for both field and lab tests. Although this reduction in exchange capacity due to residual absorption is likely minimal, we aimed to prevent unnecessary pollution. Therefore, we rinsed the resin columns three times with demineralized water, following standard laboratory cleaning procedures. Upon the third rinse with demineralized water, we sampled the columns for chemical analysis of HCl concentration, which was found to be below the detection limit. This confirmed that the columns were thoroughly cleaned. We omitted this additional verification from the manuscript for readability.

> ➢ We added the reason why we washed the empty columns with 0.2M HCl (to remove weakly attached chemicals from the column walls) and the reason why we rinsed the empty columns with demineralized water (to ensure the removal of any remaining HCl) to the manuscript text.

Line 136: Thereafter, the deposition of the summer was multiplied by 2, which is an average correction factor to convert bulk deposition to throughfall (Table S1). - It is unclear how 2 is an average of what it is shown in Table S1. Moreover, Table S1 presents factors to convert bulk to total deposition, but in the text it is stated that the factor is used to convert to throughfall deposition values.

> This is indeed unclear. We have now changed the header of table S1 to "Ratios factor of throughfall (mostly also including stemflow, SF) to bulk deposition reported in literature.

> In the manuscript text, we added a note that the factor of 2 is primarily based on the tracer Na. The other elements showed high variation in their throughfall to bulk deposition ratios, but generally, a factor of 2 is applicable for most elements. The revised Table S1 now shows the throughfall to bulk deposition data, replacing the original table.

Line 151: Three loaded resin columns were thereafter flushed with demineralized water to test the stability of the adsorption. - Which three resin columns?

> These were three of the columns that were loaded up to 70% of the resin's capacity. These columns were not used in the further lab testing, so no extraction test was performed on these columns. To clarify this, the sentence in the manuscript is changed to 'Three resin columns loaded up to 70% of the resin's exchange capacity were thereafter flushed with demineralized water to test the stability of the adsorption.'

Line 152: Both the samples of the leachate and the demineralized water used to wash the loaded columns, were analyzed for N-NH4, and N-NO2 + N-NO3 content using a Segmented Flow Analyzer (SFA type 4000, Skalar Analytical B.V., the Netherlands), and the content of Ca, Cu, Fe, Mg, Mn, Na, total-P, S and Zn using the ICP-AES (Thermo-Scientific iCAP 6500 DUO, USA). - This sentence should be moved to end of the next paragraph, adding the extracted solution to the list of samples analysed.

> The sentence was relocated to the end of the preceding paragraph, with additional details included regarding the analysis of the extracted solution. Here's the refined version:

> "The samples of the leachate of the micro- and microfluid to load the columns, of the demineralized water to wash the loaded columns, and the samples of the extraction of the elements from the columns were analyzed. Specifically, N-NH4 and N-NO2 + N-NO3 content were determined using a Segmented Flow Analyzer (SFA type 4000, Skalar Analytical B.V., the Netherlands), while the content of Ca, Cu, Fe, Mg, Mn, Na, total-P, S, and Zn was analyzed using the ICP-AES (Thermo-Scientific iCAP 6500 DUO, USA)."

> By integrating this information into the next paragraph, the original sentence at line 152 was removed.

Table S2: Please consider changing macro- and microfluid by "macro- and micro elements".

> We have updated the terminology in Table S2 from 'macro- and microfluid' to 'macro- and micro solution', which we believe more accurately represents the entire solution containing these elements, not just the elements themselves. We trust this adjustment aligns with your expectations and is suitable for the manuscript.

Table 1: Please consider changing here (and in the rest of the manuscript) extraction fluid by "extraction solution".

> ➤ We have changed the phrase "extraction fluid" to "extraction solution" throughout the manuscript, as per your suggestion.

Table 2: The columns of Paired samples (Bulk deposition and Throughfall) seem to be switched.

> ➤ You are correct. I have switched the column names so that the correct names are now above the appropriate columns.

Line 169: The extraction method was either drip, in which the extractant was slowly dripped over the resin, or a shake drip combination in which the resin was shaken in 50 mL of the extractant for 1 hour and the remaining extractant was dripped over the resin. - For the shake-drip method, how the resin was shaken? was the resin put into the column again after shaken it in a plate? Please, clarify.

> ➤ The resin was shaken using a GLF 3015 platform shaker at a speed of 120 movements per minute. To facilitate the shaking process, half of the extraction fluid was added to a clean 50 ml centrifuge tube (item No 210261, Greiner bio-one) with a blue screw cap to prevent sample loss. After shaking, the resin was returned to the original column, and the extractant in which the resin was shaken was allowed to drain from the resin and captured. This involved allowing the shaken extraction solution to drip out of the resin. Subsequently, the remaining half of the extraction fluid was allowed to drip over the resin. These details have been added to the manuscript for clarification.

Line 226: the resin columns were collected on January 14th, 2020, dried together with lab blanks to a constant weight at 28ºC and subsamples were taken for 2M KCl extraction followed by N-NH4 and NNO2 + N-NO3 content analysis using a Segmented Flow Analyzer (SFA 4000, Skalar Analytical B.V., the Netherlands) and for 3.5M HCl extraction followed by Ca, Cu, Fe, Mg, Mn, Na, P, S and Zn content analysis using the ICP-AES. - Why this methodology (drying and concentration of extraction solution) was selected for the field comparison? Also, what volume of extraction solution was used?

> ➤ We chose to dry the resin because we needed at least three samples per resin column: one for KCl extraction to measure nitrogen components, one for HCl extraction to measure other elements, and one as a spare in case of processing errors or contamination. When necessary, we used these spare samples to ensure the best data quality. Splitting the sample into three parts without drying would be problematic due to the varying wetness both within and between samples, which could lead to significant uncertainties.
> ➤ We choose to dry before extraction in the lab instead of after extraction to control the amount of resin compared to the amount of the extractant. This standardization needed the dry resin, which was tested and shown that drying the resin improved the extraction efficiency. During the extraction of the field samples, we used a volume of 100 mL, which was tested and validated in the laboratory.
> ➤ We selected 2M KCl and 3.5M HCl for extraction because these methods provided reliable results using the shake method. For KCl we used the 2M extractant as the effect of 1M on dry samples was not yet tested and we choose for shake-drip as this method was faster. For the HCl extraction, while other molarities appeared

to yield higher recovery efficiencies, each had its own pitfalls. For instance, the drip extraction on dried resin using 2M HCl generally resulted in a high overall recovery efficiency (94%, Table 3), but the recovery of phosphorus (P) was only 40%, making it unsuitable for P extraction. The shake-drip extraction method with 2M HCl showed values above 100%, indicating possible contamination and rendering this extractant unreliable, as the recovery percentages could not be accurate. We chose the 3.5M HCl extractant because it offered relatively high P recovery and overall reasonable recovery of other elements. In situations where P extraction is not needed, other methods may perform better, such as the 2M dry weight drip or the 4-2-1M dry weight drip method. To avoid unclarities in the manuscript, we added this information in line 230.

Line 265: The adsorption capacity was not influenced by the flushing of the resin with demineralized water. - I believe that this result corresponds to the text named before as "stability of absorption". It is not well explained in the methodology section and some more words to clarify the implications of the results could be added here or in the discussion.

> Yes, that is correct. To clarify why we tested this stability, we added a phrase to the methods section: "This stability needed to be tested to check if the ion exchange resin would release nutrients when exposed to (very) wet conditions." Furthermore, we revised the sentence in question to: "The adsorption capacity was not influenced by the flushing of the resin with demineralized water, indicating that the elements, once adsorbed, are not released through an excess of water such as heavy precipitation." Finally, in the discussion, we revised the sentence: "We show that IER is also able to adsorb above 99% for a range of other elements including the base cations and some micronutrients" to: "We show that IER is also able to adsorb above 99% for a range of other elements, including the base cations and some micronutrients, and that the adsorbed elements are not released in response to an excess of water such as heavy precipitation," indicating the implications of this test.

Line 272: Elemental adsorption within the resins exchange capacity was thus close to 100% for all elements except P which was underestimated under extreme conditions. - Please, consider adding to the sentence the following (or a similar one) particularization: "under the different simulated environmental condition".

> In response to your comment, we revised the sentence to: "Elemental adsorption within the resin's exchange capacity was thus close to 100% for all elements when the resin was used within its capacity, except for P, which was underestimated under the different simulated environmental conditions." This adjustment moves away from the term "extreme conditions" and accurately reflects the specific laboratory conditions that were tested.

Line 291: The average recovery efficiency was highest (90-100%) following either 2M HCl extraction or 4-2-1M HCl extraction. - The average here is per element or per extraction combination. Looking at the table it is not clear where this range is extracted from. P is not this range, neither FW method is. It is possible that it refers only to DW method per extraction combination? Please, clarify.

> There were two factors causing the indistinctness in this sentence. First, we forgot to state that this referred to the dried resin and second the use of a range can be misleading here, as we only refer to 3 values shown in

the column Avg in table 3. For clarification we changed the sentence to "The highest average recovery efficiencies were achieved with dried resin using either 2M HCl extraction or 4-2-1M HCl extraction. Specifically, the 2M HCl methods yielded average recovery efficiencies of 94% (drip) and 100% (shake-drip), while the 4-2-1M HCl method on dried resin achieved 90% recovery efficiency."

Table 5: Please consider adding any error- or bias indicator, such as mean normalized error or/and mean normalized bias. Moreover, the acronym ORG is not explained here and it is missing from the rest of the text.

➢ We added the explanation of the ORG abbreviation in the header of Table 5. This abbreviation referred to the commonly used method for bulk and throughfall deposition. The first line of the table description is now as follows: "Regression Coefficients (Intercept and Slope ± s.e.) and $R^2$ of models for the relation between the IER Method and the commonly used method, including correction for blanks and lab recovery (n = 18)"

➢ We added the mean absolute error to table 5. This mean absolute error of the regression model was calculated as: $MAE = (1/n) * \Sigma|y_i - x_i|$, where $Y_i$ indicates that $i^{th}$ observed value, $X_i$ indicates that $i^{th}$ predicted value and $N$ indicates the total number of observations

Line 335: First, the adsorption capacity of the IER when loaded within its capacity was generally high. - Consider clarifying here that the value was 70% of its capacity, as it is done at the beginning of the next paragraph.

➢ In line with your remark, I have revised the sentence to: " First, the adsorption capacity of the IER, when loaded up to 70% of its capacity as reported by the manufacturer, was generally high".

Line 440: The lower deposition estimates of P can be caused by a better adsorption of inorganic P compared to organic P to the resin. - Have you consider measuring $PO_4^{3-}$ in addition to P? Could it be a further-research objective?

➢ Yes, we considered measuring $PO_4^{3-}$ in addition to total P, but our attempts to successfully extract $PO_4^{3-}$ were unsuccessful. Following standard lab protocols at the CBLB laboratory, we measured $PO_4^{3-}$ concentration using the Segmented Flow Analyzer (SFA 4000, Skalar Analytical B.V., the Netherlands), the same device used for measuring $N-NH_4^+$ and $N-NO_3^-$ concentrations. While the device works well with salty solutions, it has issues with acidic solutions, limiting the extractants we could use for measuring $PO_4^{3-}$.

➢ We tried extracting $PO_4^{3-}$ using 2M KCl, which resulted in an average recovery of only 8% (n = 4). Due to this very low recovery rate, we focused on optimizing total P extraction with HCl, as presented in this paper, and did not pursue further optimization of $PO_4^{3-}$ extraction. Optimizing $PO_4^{3-}$ extraction remains a further-research objective, with several potential extractants still to be tested for better recovery rates.

➢ We added this to the discussion reflecting on the adsorption of inorganic P and organic P and the knowledge gap due to the problems with $PO_4^{3-}$ extraction.

Line 460: Our results even imply a higher reliability of the IER-method than the water method since uncertainties related to biological reactions and the detection limit for lab measurements could be removed. - I strongly recommend adding a clarification to this sentence, such as "under certain circumstances". In the field work of the present study, the IER funnels were cleaned weekly (if contaminated), which is something that cannot occur when collectors are only

visited seasonally (or longer). Take into account that IER method is intended also for avoiding frequent visits to field, e.g. in locations with difficult access.

> ➤ We added this clarification in the sentence and added two sentences looking into the contamination issue. Our conclusion is now rewritten to: "Our results even imply a higher reliability of the IER-method than the water method under certain circumstances since uncertainties related to biological reactions and the detection limit for lab measurements could be removed. However, possible contamination of the IER collectors due to factors such as bird feces or other animal disturbances is a point of concern, as long field exposure increases the risk of contamination. It is therefore recommended to increase the number of samplers when using the IER method. We conclude that IER is a powerful tool for the monitoring the element input by bulk deposition and throughfall for of a broad range of elements, across a broad range of environmental conditions" . We added the sentences between 'However' and 'using the IER method'.

Finally, a common concern in the use of IER is the pH of the resultant extracted solution. This can pose a problem for the analysis of some elements or their conservation in the sample. Did you perform a pH test in the extracted samples? Do you have any comment on this?

> ➤ We did not perform pH tests on the extracted samples in this study. However, we acknowledge that the pH of the resultant extracted solution is a significant consideration in ion exchange resin (IER) applications. To stabilize the samples prior to chemical analysis we diluted the KCl sample 4 times, reducing the molarity to 0.5M. The pH of the HCl extractions were likely highly acidic. A 1M HCl solution has a pH of 0, while the pH of HCl solutions with higher molarities become negative. The extracted elements from the resin will increase this pH but we can assume that the pH was still very acidic.

---

## Author Comment (AC3)

**Reviewer 1**

General Comments: Ion exchange resin precipitation collectors (IERs) have been used to quantify element/ion deposition in remote locations, but there remain questions about the reliability of IERs. This study examined different laboratory approaches (wet vs dry resins; drip vs drip-shake extraction; molarity of extraction) to processing IERs and also compared IERs to more traditional wet precipitation collectors in a forest with four different canopy covers. The study addresses important questions. However, the experimental design is a major weakness of the work. In both the laboratory and the field part of the work, complete statistical analyses are not shown. Given the unbalanced design and low replication (n=1 for some treatment combinations in the lab study), I read, but did not comment on the Discussion. I was not convinced that the laboratory study especially provided convincing new insights into questions about IER processing.

Below we answer two main aspects, i.e., low replication and experimental design versus statistical analysis.

Low replication

➢ The reviewer is correct; our sample size is indeed small. For the extraction tests conducted in the laboratory, we were limited to $N = 2-3$ due to constraints on time and budget. We acknowledge this limitation and have removed any tests conducted with only one sample from the manuscript.

➢ Duplo samples are often utilized in the early stages of research where the primary goal is to assess the feasibility of a method. This was the objective of our study. To provide insight into the variation between these samples, we have included the standard error in Table 5 (Table 3 in the original manuscript), where we report the recovery efficiency following extractions. While larger sample sizes would certainly reduce standard errors further, we noted that the standard errors tend to be lower when the recovery approaches 100%, indicating minimal differences between duplicate and triplicate samples in cases of high recovery efficiency.

➢ We have added six lines to our discussion (section 4.2) to address our low sample size and recommend larger sample sizes for future testing of the resin's recovery efficiency. The added lines are as follows: "For this test, we used generally duplicate or triplicate samples, which can be considered a low sample size. However, because these tests were performed under controlled laboratory conditions, a small sample set can be justified. When using the ion exchange resin method for field studies, we recommend testing the extraction fluid with a larger number of samples to reduce the standard error (Table 5). Nonetheless, we are confident that our conclusions are justified, given the controlled circumstances of the laboratory tests and the relatively low standard errors."

➢ In summary, while our sample sizes were small, the controlled laboratory conditions and low standard errors support the reliability of our conclusions.

Experimental design and statistical analysis

➢ Our objective was not to test differences between treatments, but rather to identify treatments that consistently yield good and reliable results for the IER method. Furthermore, we want to emphasize that (1) the adsorption

capacity, stability and recovery efficiency of the resin must be tested when using the ion exchange resin method and (2) to provide a way how to test this adsorption capacity, the stability and the recovery efficiency of the resin. In our effort to show the different ways how to extract the resin we ended up with a design with different sample sizes for the different groups. To address this, we excluded groups with only one sample and focused on treatments with two or more samples to ensure the robustness of our method's results. Furthermore, as explained in more detail by the specific comments regarding the extraction efficiency, the unbalanced design does not necessarily pose problems when using Anova as long as the type of the Sum of Squares is reported and as long as the interaction effects are significant: see Smith and Cribbie (2014) for a comprehensive comparison of unbalanced Anova using type I, II and type III sum of squares. Furthermore, with the unbalanced design of this study, we demonstrate that the IER method can be effectively employed with various extractants, resin types, and extraction equipment. We show that reliable results can be obtained using self-made equipment which makes the method well applicable all around the world.

Specific comments

Lines 107ff – IER construction is not the same across all published IER studies.  Please specify the diameter and length of the resin tubes themselves.

➢ The diameter and length of the resin tubes are indeed important to include in the manuscript. In the methods section of the original manuscript, we first described the preparation of the resin columns before we described the deposition samplers in more detail, thus including the requested information in line 198. In the revised manuscript, we added this information right after line 107. The revised text is as follows: "We prepared 45 resin columns for the laboratory tests of elemental adsorption and recovery (including the blanks), followed by the preparation of 30 column's for the field test of the IER-method. The resin columns had a volume of 15.7ml and an inner diameter and length of respectively 12.4 and 130mm. First, the resin column's were cleaned using 0.2M HCl and demineralized water to remove weakly attached chemicals from the column walls."

Line 115. Change weighted to weighed.

➢ We changed this word, comment was followed.

Line 127 – Existing deposition data for where?

➢ The deposition data was taken from nearby measurement stations. The locations of these measurement stations are now included in line 127-128. We used the data of 2015 from all these measurement stations in which the wet-only deposition is sampled weekly or biweekly. We were not able to use more recent data as these were not yet available.

Lines 128-135.  This is a long sentence – and one that I do not fully understand.  "…representative stations" – representative of what?  What does "for the funnel surface" mean?

➢ The phrase 'representative stations' is changed to 'the nearby weather stations', which are now also explicitly mentioned. Line 128-129 is now changed into: "First, based on existing wet-deposition data from nearby measurement stations located in Biest Houtakker, Speuld, De Zilk and Vredepeel (NL) (RIVM, 2015), we estimated the bulk deposition amounts (kg ha$^{-1}$) for different elements, and then used those to determine the needed molarity of the solution that was used to test the adsorption capacity of the resin".

➢ With the funnels surface we meant that the seasonal concentrations in the wet-deposition, which were expressed in mg L$^{-1}$, were multiplied by the precipitation that would be captured by the funnel (in L) by multiplying the recorded precipitation (in mm or L m$^{-2}$) with the horizontal surface of the funnel (in m$^2$).We now removed the phrase "funnels surface" and split the sentence in three different sentences for better understandability: "To estimate the maximum bulk deposition values, the monthly measurements of existing bulk deposition data of the nearby weather stations (umol l$^{-1}$) (RIVM, 2015), were summed to seasonal concentrations, expressed in mg L$^{-1}$. Then, the stations were selected with the highest seasonal deposition, occurring during summer, for both macro- and micronutrients, based on the total molarity of the rainwater. These seasonal concentrations were then multiplied by the precipitation (in L) that would be captured by the funnel, by multiplying the recorded precipitation (in mm or L m$^{-2}$) with the horizontal surface of the funnel (in m$^2$) to estimate the total deposition captured by a funnel.".

Lines 135-136 – Table S1 is referenced in text, but when I read Table S1 I had a hard time understanding what the Table was showing or how it related to the sentence in the manuscript.

➢ We agree that the heading of table 1 was confusing as we suggested that the reported values were used to calculate the total deposition instead of the throughfall. In Table S1 we presented "Ratios of throughfall (mostly also including stemflow, SF) to bulk deposition reported in literature" as now clearly mentioned in the table heading. We used this information to derive an average multiplication factor of 2 to convert bulk deposition to throughfall, based on the reported values for the ratio throughfall/bulk deposition of the tracer Na. This has now been rephrased in this way in the main text (Table S1).

Lines 138-142.This information could be put in to a small table. It would be useful to give the recipe for these solutions (how many grams of each salt?).

➢ We placed the information of line 138 to 142 in a new table (shown below). Therefore, we changed the text in line 138 to: "The total elemental content of this throughfall flux, multiplied by 4 (assumed that the summer values are representative of the entire year, which is a precautionary approach), was dissolved in a 1 L solution separately for macro and micronutrients using stock solutions resulting in an extraction solution containing values reflecting the maximum annual total deposition in the Netherlands (Table 1)" .

Table 1: The throughfall flux used to test the adsorption capacity and recovery efficiency of the ion exchange resin. We used stock solutions with known molarity to make the macro and the micro solution used to drip through the resin. The total volume of the used stock solution (in ml L$^{-1}$) and the concentration in umol per element are given.

| Stock solution Code | Mol | Type | Total mL L$^{-1}$ | Ca | Cu | Cl | Fe | K | Mg | Mn | Na | PO$_4$ | SO$_4$ | Zn | NH$_4$ | NO$_3$ |
|---|---|---|---|---|---|---|---|---|---|---|---|---|---|---|---|---|
| | | | | | | | | | *umol* | | | | | | | |
| Na$_2$SO$_4$ | 0.5 | Macro | 0.90 | | | | | | | | 450 | | 450 | | | |
| NaCl | 1 | Macro | 1.40 | | | 1400 | | | | | 1400 | | | | | |
| KNO$_3$ | 1 | Macro | 0.18 | | | | | 180 | | | | | | | | 180 |
| KH$_2$PO$_4$ | 1 | Macro | 0.02 | | | | | 20 | | | | 20 | | | | |
| NH$_4$NO$_3$ | 1 | Macro | 1.82 | | | | | | | | | | | | 1820 | 1820 |
| NH$_4$Cl | 1 | Macro | 2.18 | | | 2180 | | | | | | | | | 2180 | |
| MgSO$_4$ | 1 | Macro | 0.3 | | | | | | 300 | | | | 300 | | | |
| CaCl$_2$ | 0.5 | Macro | 0.8 | 400 | | 400 | | | | | | | | | | |
| FeCl$_2$ | 0.1 | Micro | 6.0 | | | 600 | 600 | | | | | | | | | |
| Cu(NO$_3$)2.3H$_2$O | 0.275 | Micro | 0.036 | | 9.9 | | | | | | | | | | | 9.9 |
| Zn(NO$_3$)2.6H$_2$O | 0.267 | Micro | 0.075 | | | | | | | | | | | 20 | | 20 |
| Mn.SO$_4$.H$_2$O | 0.01 | Micro | 15.0 | | | | | | | 150 | | | 150 | | | |
| | | | **total** | 400 | 9.9 | 4580 | 600 | 200 | 300 | 150 | 1850 | 20 | 900 | 20 | 4000 | 2030 |

Line 145 – Should be column or column's, not columns

➤ Changed.

Line 147 – Should be resin or resin's, not resins

➤ Changed.

Lines 150-155 – Were the leachate and demineralized water samples filtered prior to analysis? Hopefully, yes.

➤ Filtering water samples before chemical analysis is typically done for four main reasons: removing particles, preventing contamination, ensuring accuracy and precision, and protecting analytical instruments. However, in this specific laboratory test, filtering was not necessary for several reasons. First, the resin was thoroughly flushed (500 grams of resin flushed with 8 liters of demineralized water), ensuring no small particles remained. Additionally, all materials that came into contact with the nutrient-containing fluids, which simulate total deposition, were cleaned according to standard laboratory protocols, minimizing the risk of contamination from large particles. The nutrient solution, representing the total annual deposition, was well mixed, with no undissolved particles visible. Therefore, filtering was not needed to prevent contamination or protect analytical instruments. Importantly, unnecessary filtering can introduce contaminants, potentially reducing the accuracy and precision of the analysis. To maintain sample integrity, filtering was avoided unless absolutely necessary which was not the case for this study. Finally, the accuracy and precision of the analysis

were validated by including standard samples with known concentrations, a routine procedure in the lab. This ensured the reliability and accuracy of the results without the need for additional filtering.

Line 163 – "…previous studies…" but only one reference. I do not see in Fenn et al. (2018) a discussion of the molarity of the extraction solution. That paper does have a section of using solutions other than KCl or KI is a researcher wants to quantify K+ deposition.

> This source did indeed not justify our claim. We changed the text of the manuscript to "since a higher recovery of the base cations was found with a 1M HCl extraction compared to a 0.5M HCl extraction". This can be found on page 53 of Fenn et al., 2018.

Table 1 – These are very low sample sizes.

> This point has been answered in our reply to the main comment on low replication.

Line 185 - Table 2 – I think I can see why there is not an equal number of samplers in bulk deposition and throughfall in each treatment, but statistically, I do not see how the design presented in Table 2 would work.

> In response to the reviewers comments, we changed the statistics, now using another aspect of the design. Initially we compared the IER-method and the original water method for each harvest intensity treatment thereby ignoring the funnels placement. The harvest intensity treatment is shown in table 2. This test was set up to result in 4 groups containing 7 paired samplers. However, upon reflection, we realized our main focus was comparing the water-based deposition method with the ion exchange resin method in forest gaps (bulk deposition) and beneath the canopy (throughfall). To clarify, we updated the Table 2 header to specify that we only tested both methods in throughfall and bulk deposition. The original design included 13 paired throughfall samplers and 15 paired bulk deposition samplers, but due to bird feces contamination, we ended up with 9 pairs of throughfall samples and 9 pairs of bulk deposition samples. Initially, we reported different treatments expecting higher deposition in small forest gaps, influenced by surrounding trees. However, this expectation is only relevant if the IER method performs differently than the water method for throughfall and bulk deposition which was not the case (see revised version of figure 4). If no such difference exists, there's no need to control for gap size in comparing the IER method to the water method. Our results indicate that IER and water method covary consistently independent of forest gap versus crown cover (bulk vs. throughfall).

> In line with this, we revised the field study statistics to exclude the treatments (control, high-thinning, shelterwood, and clearcut), focusing instead on the funnel's position (throughfall or bulk deposition). We ran linear models for all elements with the funnel position as a random structure. These models were constructed as $y_{ij}=\beta_0+\beta_1 x_{ij}+u_j+\epsilon_{ij}$ , where $y$ is the result using the IER method, $x$ is the result using the original method, $u$ represents the random structure, and $\epsilon$ the residuals. The random structure was included only if it improved model performance by $\Delta 2$, following Zuur et al. (2009). None of the models showed improved performance

with the funnel position as a random variable. The results are visualized in the revised Figure 4. The statistics are added to section 2.4 of the revised manuscript.

Line 192 – What is a "common" deposition collector?

➢ This are the bulk and throughfall deposition samplers that are commonly used in other studies as well (Bleeker et al., 2003) and are designed to capture and store the precipitation. This precipitation is then sampled for volume and nutrient concentration to calculate the bulk or throughfall deposition. To clarify this, we added the phrase 'collecting the precipitation next to and below the forest' between brackets right after mentioning this common deposition collector.

Line 193 – Maybe add "collectively" after "The 7 collectors per plot"

➢ Added

Line 222 – Change send to sent

➢ Changed

Lines 222ff – Were the samples filtered?

➢ Yes, the samples of the original water method were classified as surface waters and are therefore filtered following the standard procedure of the CBLB laboratory. We now added this.

➢ The samples of the IER-method were not filtered as explained in the reply to the comments on line 150-155. We added this information.

Line 223 – Change contents to concentrations

➢ Changed.

Lines 246-254 – Calculating adsorption capacity and recovery efficiency based on analyte concentrations assumes that the volume of the added solution is the same as the volume of the extract solution. Was this always the case?

➢ The volume of the added solution matched the extract solution's volume because we let the fluid fully drain from the resin. Thus, the subsample taken from this extractant is assumed to represent the entire extract solution. This assumption is valid as the resin was wet and gravity-drained before adding the nutrient solution, representing the total annual deposition, making the change in wetness negligible.

Lines – 263ff – The results section suffers from not considering results in light of the ANOVA and subsequent Tukey's tests. As an example, statistical results are not considered at all regarding the results shown in Figure 3. And how can it be that overloading of the cation and/or anion exchange capacity results in 100% adsorption capacity?

The reviewer addresses two points in this remark, namely the statistical tests for the adsorption capacity and the 100% adsorption for some elements when the resin is loaded beyond its capacity.

- ➢ We indeed did no statistical testing for differences in adsorption capacity since the small sample sizes strongly limit the statistical power of such tests.

- ➢ In our revised manuscript, we added a new Table 4 to replace Figure 3. In this table, we report the mean and the standard error of the mean. These standard errors are generally quite small, indicating a high precision of our measurements. Note that unlike the adsorption tests; we used ANOVA for the recovery efficiency and field tests (lines 255-259) since these data showed more variability (larger standard errors) and was less intuitive to understand. In response to your feedback, we conducted a generalized test per treatment group for the adsorption data, as individual element-specific tests were impractical due to low sample sizes and generally low standard errors, resulting in essentially constant data that lacks statistical evaluability. For elements where mean adsorption was less than 100%, we used the Wilcoxon signed rank test to assess the hypothesis of adsorption equality to 100%. To address ties (values equal to 100%), we implemented appropriate corrections within the Wilcoxon test methodology. However, we believe that the differences between the tests are readily apparent when comparing the means and standard errors of the data. Given the very small errors, we consider the observed differences as real, and not as a result of random variation.

- ➢ Overloading the cation and/or anion exchange capacity did not result in 100% adsorption capacity for all elements. We observed that only certain elements, which strongly bind to the resin, achieved (nearly) 100% adsorption capacity when the resin was overloaded beyond its exchange capacity. This suggests that the resin has a high affinity for these specific elements. In contrast, other elements, which have lower affinity for the resin, showed lower adsorption capacity under similar conditions. We discuss this phenomenon in detail in the manuscript, specifically in lines 349–362. For clarity we add these lines here: "To further test the affinity of the resin for the studied elements, the resin was loaded to approximately 160% and 240% of its capacity. Based on the adsorption capacity beyond the resins capacity, we found that the cation bed has an affinity of $Ca = Fe > Cu = Mn = Zn > Mg > K > NH_4 > Na$ which is in line with the previous reported resin affinity (Skogley and Dobermann, 1996). The anion bed has an affinity of $S > NO_3 > P$ which agrees with earlier studies (Skogley and Dobermann, 1996; Park et al., 2014). The resins affinity and the adsorption capacity for different levels of loading beyond the resins capacity is of importance for resin columns under suspicion of overloading. We did not find lower adsorption of Ca and Fe and only slightly lower adsorption of Cu, Mg, Mn and Zn, indicating that, when columns are slightly overloaded, these estimates are still reliable. When columns are loaded > 100% of the capacity, the estimates for K, Na, P, S, $NH_4$ and $NO_3$ are not reliable. Therefore, in case of suspicion of ion exchange overload, tests are recommended to check if stoichiometry between any element of Ca, Cu, Mg, Mn and Zn with K, Na, P, S, $NH_4$ and $NO_3$ falls within the stochiometric range of natural deposition estimates. We strongly recommend collecting the resin columns prior to resin saturation as adsorption of Na and P can further decrease when saturating the resin up to 90 or 100%. The time period that the resin can stay in the field depends on the total atmospheric deposition and the volume of resin used. For remote areas with low deposition levels and low risk of sample contamination (e.g. by bird feces) the resin can stay for multiple months up to a year in the field as long as adequate resin volumes are used".

Lines 290ff – Table 3 does not show statistical interactions. The bottom row shows mean recovery efficiencies for each element, but there were element interactions with pretreatment, molarity, and extraction type, so averaging element means across all treatments ignores the ANOVA interactions. Table 3 clearly illustrated the unbalanced ANOVA design and exceptionally minimal replication. Table S3 shows part of an ANOVA, but it only shows 2-way interactions (and not all of the 2-way interactions). A complete ANOVA would have 3-way interactions and one 4-way interactions. It is hard to tell what complete ANOVA would look like (it might not even run because of the unbalanced design with low replication).

➢ The reviewer addresses here multiple concerns. We have considered all these concerns point by point:

o Table 3 (table 5 in the revised version) indeed does not show statistical interactions. We provided the arithmetic means and the arithmetic average of these means. To clarify this, we changed the average recovery to average arithmetic recovery to clearly indicate that we chose to give the arithmetic means here. The reviewer is right that it is necessary to give an overview of the interaction effects. The effects of the ANOVA interactions, following a Tukey's post-hoc test, are now added to the supplements in table S6-S9. The ANOVA test results are still in table S3.

o Table 3 (table 5 in the revised version) indeed show the unbalanced design consisting of duplo measurements and one group with 6 samples. We now removed the groups who had only one sample as these groups did not add to the research output of this paper and caused confusion, especially regarding the statistics. The choice for duplo measurements is already argued in the reply to the previous comment regarding table 1. The unbalanced design does not necessarily pose problems when using Anova as long as the type of the Sum of Squares is reported and as long as the interaction effects are significant: see Smith and Cribbie (2014) for a comprehensive comparison of unbalanced Anova using type I, II and type III sum of squares.

o In Table 3 (table 5 in the revised version), we presented the complete ANOVA results, which include the elements, pre-treatment, molarity, and extraction type as explanatory variables, along with all two-way interactions involving the elements. However, we cannot test for interactions between pre-treatment and molarity or between pre-treatment and extraction type because we did not test the two extraction types for each molarity, nor did we test the pre-treatment for each molarity. We did not pursue a full factorial design involving molarity, pre-treatment, and extraction type because we believe the effect of pre-treatment (drying the resin prior to extraction) does not vary with different molarities of the extractant. Additionally, while the extraction method (drip versus shake-drip) could affect the contact time of the fluid with the resin and potentially interact with different molarities, our goal was not to determine the specific molarity at which the shake-drip method becomes superior to the drip method. Because we had the data, we expanded the Anova and now added the interaction between the pre-treatment and the extraction type in the revised table S3.

o Finally, we intentionally did not include three- and four-way interactions in our analysis. These higher-order interactions significantly increase the complexity of the analysis and can be difficult to interpret. In this manuscript, our goal is to identify an appropriate extraction method for the ion exchange resin method using tested adsorption and recovery percentages. We are not focused on

specific combinations of molarity, pre-treatment, and extraction type that may lead to higher recovery percentages. Our aim is to determine general patterns, such as whether increasing molarities is beneficial or whether a shake-drip method is preferable to a drip method. While a four-way interaction might reveal that a specific combination works best for a particular element, this does not necessarily improve the overall estimates of deposition using the IER method, as long as the adsorption and recovery percentages are reliable and have a small standard error. Reliable estimates for adsorption and recovery are far more critical than finding the perfect extraction method. Including all possible interaction terms can lead to overfitting, capturing noise rather than true underlying relationships. It also increases the risk of multicollinearity and, in cases of sparse data, can result in unreliable and unstable estimates of interaction effects. We believe in maintaining simplicity in statistical models while adequately explaining the data. Including unnecessary higher-order interactions violates the principle of parsimony and can obscure the main effects and lower-order interactions that are more important and interpretable.

Lines 310 ff – In Figure 4 and Table 5, if the goal is to compare corrected, blank corrected and recovery corrected regressions, it would be appropriate to use ANCOVA for homogeneity of slopes. This would be a better approach than simply showing the highest R2 value in bold. One cannot compare log-transformed regression coefficients to non-transformed regression coefficients (for S).

➢ In figure 4, the goal is to compare the data from the IER collectors and the water collectors using the adsorption capacity and the recovery efficiency for the different elements as determined with the laboratory test. Therefore, there is no need to change the statistics here as we do not want to compare corrected, blank corrected and recovery corrected regressions in this figure.

➢ We indeed aim to compare the corrected, blank corrected, and recovery corrected regressions in Table 5. However, we believe that using ANCOVA for homogeneity of slopes is not appropriate for this situation because we do not have separate blanks for each column nor separate recovery efficiencies for each column. This forces us to use the average contamination in blanks and the average recovery efficiency as a covariate variable in comparison to the uncorrected data of the IER collectors. ANCOVA requires the covariate to vary among the individuals in the dataset. A single average value for the covariate is not suitable and will not provide meaningful adjustments in the analysis. Without variability, a single value cannot explain any variance in the dependent variable. Including a covariate that is constant across all observations is redundant and does not contribute to the model, as it fails to adjust the dependent variable based on individual differences. Furthermore, it is not possible to use the uncorrected IER data as a covariate for the corrected IER data as an ANCOVA as there should be perfect correlation between the independent variables. As the IER data can only be corrected by averages, the covariate and the independent variable will be perfectly correlated violating the ANCOVA assumptions making this test not suitable for this data.

➢ It is indeed true that log-transformed regressions cannot be compared to non-transformed regression coefficients. We have revised the complete statistics in response to your comment on Table 2. This revision involved adding the funnel position (indicating bulk deposition or throughfall deposition) as a random factor,

but only if it would improve the regression model AIC by $\Delta 2$. Similar to the models shown in Figure 4, this was not the case for the IER data, whether corrected for blanks or for recovery efficiency. In the revised statistics, we have now moved away from using data transformation, resulting in comparable models.

➢ Finally, in order to increase the comparability between the models, we added the mean absolute error of each model to table 5.

Line 326 – I do not see that Fig. S1 shows canopy openness results.

➢ The canopy openness treatment related to the forest harvest intensity treatments as shown in table 2. In response to the comments on table 2 we changed the entire statistics and moved away from these canopy openness treatments (consisting of the control forest, thinned forest, shelterwood cut and clearcut) and focused now only on the throughfall and bulk deposition types. We therefore changed line 326 and removed the statement regarding canopy openness results.

**Literature**

Bleeker, A., Draaijers, G., van der Veen, D., Erisman, J.W., Mols, H., Fonteijn, P., Geusebroek, M., 2003. Field intercomparison of throughfall measurements performed within the framework of the Pan European intensive monitoring program of EU/ICP Forest. Environ Pollut 125, 123-138.
Park, S.-C., Cho, H.-R., Lee, J.-H., Yang, H.-Y., YANG, O.-B., 2014. A study on adsorption and desorption behaviors of 14C from a mixed bed resin. Nuclear Engineering and Technology 46, 847-856.
Skogley, E.O., Dobermann, A., 1996. Synthetic ion-exchange resins: Soil and environmental studies. J Environ Qual 25, 13-24.
Smith, C.E., Cribbie, R., 2014. Factorial ANOVA with unbalanced data: a fresh look at the types of sums of squares. Journal of Data Science 12, 385-403.
Zuur, A., Ieno, E.N., Walker, N., Saveliev, A.A., Smith, G.M., 2009. Mixed effects models and extensions in ecology with R. Springer Science & Business Media.

---

## Author Response (AR2)

Replies to reviews on Testing Ion Exchange Resin for quantifying bulk and throughfall deposition of macro and micro-elements on forests by Vos, Marleen A.E, et al.

**Reply to anonymous referee #1**

Overall, the authors have done their best to address the comments of all reviewers, but I still find sample sizes of 2 for the lab work to be problematic.

We acknowledge that the small sample size is a limitation and agree that further testing of the ion exchange method for deposition measurements is necessary. However, as mentioned in our response to major revisions, we remain confident in the reliability of our findings. We hope this paper encourages others to thoroughly test and refine the method before full-scale implementation, as not all studies using this method have tested the recovery and adsorption before implementing it.

Lines 52-53 – Please use subscripts for NH3 and NOx; and throughout, please show the charges for NH4 and NO3.

We checked the entire manuscript for the use of subscripts and used them consequently. We also added charges for $NH_4^+$ and $NO3^-$.

Lines 61-2 – Maybe I missed it, but I didn't see that Fenn and Poth 2004 claims that resins inhibits mineralization, nitrification and denitrification.

You are correct. In Fenn and Poth (2004), there is a discussion on whether differences in $NH_4^+$ concentrations can result from nitrification processes, which are primarily associated with common water collectors. Based on their discussion, we concluded that (1) nitrification does not occur in resin columns, unlike in the common water method, and (2) the higher NH4+ concentrations observed in resin columns could be due to the release of amino groups. These points are covered in the section "Possible Sources of Ammonium Discrepancy between Collector Types" in Fenn and Poth (2004).

Since Fenn and Poth (2004) discuss but do not definitively prove the inhibition of nitrification in resin compared to the water method, we revised lines 61-62 to: "Furthermore, the method is regarded as potentially more reliable for nitrogen, as the resin likely inhibits mineralization, nitrification, and denitrification, which can be affected by local weather conditions, as discussed by Fenn and Poth (2004) and Kohler et al. (2012).

Lines 65-67 – I'd suggest deleting the final sentence of this paragraph. It doesn't seem very relevant to the work at hand.

We deleted the final sentence of the paragraph.

Line 84 – The Qian and Schoenau (2002) reference is about using IERs to assess nutrient availability in soil; the Bayer et al. (2012) reference is about using IERs to remove Zn from wastewater. I'm not sure that these references are relevant to the work at hand.

We find these references valid as there are limited studies that specifically test the behavior of ion exchange resin (IER) under various field conditions. Although the resin is applied differently in these studies, the underlying principle remains consistent—the resin adsorbs elements passing through it. Given the lack of a broad spectrum of literature on the resin's behavior in environmental contexts, and considering the comparable function of the resin across different methods, we prefer to retain these references in the manuscript.

Lines 175-176 – "…since a higher recovery of the base cations was found with a 1M HCl extraction compared to an 0.5M HCl extraction (Fenn et al., 2018)" suggests that Fenn et al. experimentally compared the two HCl strengths, which is not the case. What the Fenn et al. (2018) paper says is: "Yamashita et al. (2014) used 0.5 N HCl as an extractant for SO42- and base cations, but we found better recovery of base cations using 1 n HCl. I'd suggest rephrasing or deleting this.

We rephrased the sentence to 'since a higher recovery of the base cations was found with a 1M HCl extraction

(Fenn et al., 2018) compared to a 0.5 M HCl extraction (Yamashita et al., 2014)'.

Line 180 – Suggest deleting "respectively".

We deleted the word "respectively" in this sentence.

Line 192-193 - Change content to concentrations.

Changed.

**Literature**

Fenn, M.E., Bytnerowicz, A., Schilling, S.L., 2018. Passive monitoring techniques for evaluating atmospheric ozone and nitrogen exposure and deposition to California ecosystems. Gen. Tech. Rep. PSW-GTR-257. Albany, CA: US Department of Agriculture, Forest Service, Pacific Southwest Research Station 257.
Yamashita, N., Sase, H., Kobayashi, R., Leong, K.-P., Hanapi, J.M., Uchiyama, S., Urban, S., Toh, Y.Y., Muhamad, M., Gidiman, J., 2014. Atmospheric deposition versus rock weathering in the control of streamwater chemistry in a tropical rain-forest catchment in Malaysian Borneo. Journal of tropical ecology 30, 481-492.